# Multiple-policy Evaluation via Density Estimation

**Yilei Chen** [1]   **Aldo Pacchiano** [1 2]   **Ioannis Ch. Paschalidis** [1]

## Abstract

We study the multiple-policy evaluation problem where we are given a set of $K$ policies and the goal is to evaluate their performance (expected total reward over a fixed horizon) to an accuracy $\epsilon$ with probability at least $1 - \delta$. We propose an algorithm named CAESAR for this problem. Our approach is based on computing an approximately optimal sampling distribution and using the data sampled from it to perform the simultaneous estimation of the policy values. CAESAR has two phases. In the first phase, we produce coarse estimates of the visitation distributions of the target policies at a low order sample complexity rate that scales with $\tilde{O}(\frac{1}{\epsilon})$. In the second phase, we approximate the optimal sampling distribution and compute the importance weighting ratios for all target policies by minimizing a step-wise quadratic loss function inspired by the DualDICE (Nachum et al., 2019) objective. Up to low order and logarithmic terms CAESAR achieves a sample complexity $\tilde{O}\left(\frac{H^4}{\epsilon^2} \sum_{h=1}^{H} \max_{k \in [K]} \sum_{s,a} \frac{(d_h^{\pi^k}(s,a))^2}{\mu_h^*(s,a)}\right)$, where $d^\pi$ is the visitation distribution of policy $\pi$, $\mu^*$ is the optimal sampling distribution, and $H$ is the horizon.

## 1. Introduction

This paper delves into the problem of policy evaluation, a fundamental problem in RL (Sutton and Barto, 2018) of which the goal is to estimate the expected total rewards of a given policy. This process serves as an integral component in various RL methodologies, such as policy iteration and policy gradient approaches (Sutton et al., 1999), wherein the current policy undergoes evaluation followed by potential updates. Policy evaluation is also paramount in scenarios where prior to deploying a trained policy, thorough evaluation is imperative to ensure its safety and efficacy.

Broadly speaking there exist two scenarios where the problem of policy evaluation has been considered, known as *online* and *offline* data regimes. In online scenarios, a learner is interacting sequentially with the environment and is tasked with using its online deployments to collect helpful data for policy evaluation. The simplest method for online policy evaluation is Monte-Carlo estimation (Fonteneau et al., 2013). One can collect multiple trajectories by following the target policy, and use the empirical mean of the rewards as the estimator. These on-policy methods typically require executing the policy we want to estimate which may be unpractical or dangerous in many cases. For example, in a medical treatment scenario, implementing an untrustworthy policy can cause unfortunate consequences (Thapa et al., 2005). In these cases, offline policy evaluation may be preferable.

In the offline case, the learner has access to a batch of data and is tasked with using it in the best way possible to estimate the value of a target policy. Many works focus on this problem leveraging various techniques, such as importance-sampling, model-based estimation, and doubly-robust estimators (Yin and Wang, 2020; Jiang and Li, 2016; Yin et al., 2021; Xie et al., 2019; Li et al., 2015). In the context of offline evaluation, the theoretical guarantees depend on the overlap between the offline data distribution and the visitations of the evaluated policy (Xie et al., 2019; Yin and Wang, 2020; Duan et al., 2020). These coverage conditions, which ensure that the data logging distribution (Xie et al., 2022) adequately covers the state space, are typically captured by the ratio between the densities corresponding to the offline data distribution and the policy to evaluate, also known as importance ratios.

Motivated by applications where one has multiple policies to evaluate, e.g., multiple policies trained using different hyperparameters, Dann et al. (2023) considered multiple-policy evaluation which aims to estimate the performance of a set of $K$ target policies instead of a single one. At first glance, multiple-policy evaluation does not pose challenges beyond single-policy evaluation since one can always use $K$ instances of single-policy evaluation to evaluate the $K$ policies. However, since the sample complexity of this

[1]Boston University, Boston, USA [2]Broad Institute of MIT and Harvard, Cambridge, USA. Correspondence to: Yilei Chen <ylchen9@bu.edu>.

*Proceedings of the 42nd International Conference on Machine Learning*, Vancouver, Canada. PMLR 267, 2025. Copyright 2025 by the author(s).

procedure scales linearly as a function of $K$, this can be extremely sample-inefficient as it neglects potential similarities among the $K$ target policies.

Dann et al. (2023) proposed an online algorithm that leverages the similarity among target policies based on an idea related to trajectory synthesis (Wang et al., 2020). The basic technique is that if more than one policy take the same action at a certain state, then only one sample is needed at that state which can be reused to synthesize trajectories for these policies. Their algorithm achieves an instance-dependent sample complexity which gives much better results when target policies have many overlaps while it requires estimation of generative models as an intermediate step which can be unpractical.

Different from Dann et al. (2023), in this work, we tackle multiple-policy evaluation problem from the offline perspective[1]. In the context of single-policy evaluation, an offline dataset is typically given and assumed to provide good coverage over the state space of the target policy. Nevertheless, in our scenario, such a dataset does not exist. To overcome this issue, we design our algorithm based on the idea of firstly calculating a behavior distribution with enough coverage of the target policy set. Once this distribution is computed, independently and identically distributed (i.i.d.) samples from the behavior distribution can be used to estimate the value of the target policies using ideas inspired in the offline policy optimization literature. Briefly speaking, our algorithms consist of two phases:

1. Build coarse estimators of the policy visitation distributions and use them to compute a mixture policy that achieves a low visitation ratio with respect to all $K$ policies to evaluate.

2. Sample from this approximately optimal mixture policy and use these to construct mean reward estimators for all $K$ policies by importance weighting.

Coarse estimation refers to estimating a target value up to constant multiplicative accuracy (see Definition 4.1). We show that coarse estimation of the visitation distributions can be achieved at a cost that scales linearly, instead of quadratically with the inverse of the accuracy parameter. This ensures that the sample cost in the first phase of our algorithm is of lower order and can therefore be considered negligible (see Section 4.1). We then show that estimating the policy visitation distributions up to multiplicative accuracy is enough to find an approximately optimal behavior distribution that minimizes the maximum visitation ratio among all policies to estimate (see Section 4.2). In

the second phase, the samples generated from this behavior distribution are used to estimate the target policy values via importance weighting. Since the importance weights are not known to sufficient accuracy, we propose the IDES or *Importance Density EStimation* algorithm (see Algorithm 1) for estimating these distribution ratios by minimizing a series of loss functions inspired by the DualDICE (Nachum et al., 2019) method (see Section 4.3). Combining these steps we arrive at our main algorithm (CAESAR) or *Coarse and Adaptive EStimation with Approximate Reweighing* algorithm (see Algorithm 2) that achieves a high probability finite sample complexity for the problem of multi-policy evaluation.

We summarize our contributions as the following,

- We propose a novel, sample-efficient algorithm, CAESAR , for the multiple-policy evaluation problem which achieves a non-asymptotic, instance-dependent sample complexity. CAESAR provides new results and valuable insights to the existing literature while sharing several advantages compared to existing approaches (see Section 5).

- We introduce the technique of coarse estimation and demonstrate its effectiveness in solving the multiple-policy evaluation problem. We believe this technique has potential applications beyond the scope of this work.

- We propose an algorithm, IDES , as a subroutine of CAESAR to accurately estimate the marginal importance ratios by minimizing a carefully designed stepwise loss function. IDES is a non-trivial extension of DualDICE to finite-horizon Markov Decision Processes (MDPs).

- We propose a novel notion termed $\beta$-distance along with an algorithm MARCH , which achieves coarse estimation of the visitation distribution for all deterministic policies, with a sample complexity scales polynomially in parameters such as the size of the state and action spaces, despite the fact of exponential number of deterministic policies.

## 2. Related Work

There is a rich family of off-policy estimators for policy evaluation (Liu et al., 2018; Jiang and Li, 2016; Dai et al., 2020; Feng et al., 2021; Jiang and Huang, 2020). But none of them is effective in our setting. Importance-sampling is a simple and popular method for off-policy evaluation but suffers exponential variance in the horizon (Liu et al., 2018). Marginalized importance-sampling has been proposed to get rid of the exponential variance. However, existing works all focus on function approximations which

---

[1]We say offline here to emphasize that the final evaluation is conducted in an offline fashion rather than implying there is no interaction with the environment.

only produce approximately correct estimators (Dai et al., 2020) or are designed for the infinite-horizon case (Feng et al., 2021). The doubly robust estimator (Jiang and Li, 2016; Hanna et al., 2017; Farajtabar et al., 2018) also solves the exponential variance problem, but no finite sample result is available. Our algorithm is based on marginalized importance-sampling and addresses the above limitations in the sense that it provides non-asymptotic sample complexity results and works for finite-horizon *Markov Decision Processes (MDPs)*.

Another popular estimator is called model-based estimator, which evaluates the policy by estimating the transition function of the environment (Dann et al., 2019; Zanette and Brunskill, 2019). Yin and Wang (2020) provide a similar sample complexity to our results. However, there are some significant differences between their result and ours. First, our sampling distribution; calculated based on the coarse distribution estimator, is optimal. Second, our sample complexity is non-asymptotic while their result is asymptotic. Third, the true distributions appearing in our sample complexity can be replaced by known distribution estimators without inducing additional costs, that is, we can provide a known sample complexity while their result is always unknown since we do not know the true visitation distributions of target policies.

The work that most aligns with ours is Dann et al. (2023), which proposed an on-policy algorithm based on the idea of trajectory synthesis. The authors propose the first instance-dependent sample complexity analysis of the multiple-policy evaluation problem. Different from their work, our algorithm uses off-policy evaluation based on importance-weighting and achieves a better sample complexity with simpler techniques and analysis. Concurrently, Russo and Pacchiano (2025) studied the multiple-policy evaluation problem in discounted settings from the lower bound perspective.

Analogous to our two-stage pipeline, Amortila et al. (2024) proposed an exploration objective for downstream reward maximization, similar to our goal of computing an optimal sampling distribution. However, our approximate objective, based on coarse estimation is easier to solve, which is a significant contribution while they need layer-by-layer induction. They also introduced a loss function to estimate ratios, similar to how we estimate the importance densities. However, our ratios are defined differently from theirs which require distinct techniques.

Several existing works have utilized estimates of visitation distributions up to multiplicative constant accuracy (Zhang and Zanette, 2023; Li et al., 2023). We formally formulate the technique of coarse estimation and present clean results that are ready to use in general scenarios while the existing results have more additional complex terms and are limited to the specific tasks. Moreover, our coarse estimation results are based on simple concentration inequalities and the multiplicative constant can be any value, leading to more flexible and elegant formulations.

Our algorithm also uses some techniques modified from other works which we summarize here. DualDICE is a technique for estimating distribution ratios by minimizing some loss functions proposed by (Nachum et al., 2019). We build on this idea and make some modifications to meet the need in our setting. Besides, we utilize stochastic gradient descent algorithms and their convergence rate for strongly-convex and smooth functions from the optimization literature (Hazan and Kale, 2011). Finally, we adopt the Median of Means estimator (Minsker, 2023) to convert in-expectation results to high-probability results.

## 3. Preliminaries

**Notations** We denote the set $\{1, 2, \ldots, N\}$ by $[N]$. $\{X_n\}_{n=1}^N$ represents the set $\{X_1, X_2, \ldots, X_N\}$. $\mathbb{E}_\pi[\cdot]$ denotes the expectation over the trajectories induced by policy $\pi$ while $\mathbb{E}_\mu[\cdot]$ represents the expectation over the trajectories sampled from distribution $\mu$. $\tilde{O}(\cdot)$ hides constants, logarithmic and lower-order terms. We use $\mathbb{V}[X]$ to represent the variance of random variable $X$. $\Pi_{det}$ is the set of all deterministic policies and $\text{conv}(\mathcal{X})$ represents the convex hull of the set $\mathcal{X}$. If not explicitly specified, $\forall s, a, h, k$ stands for $\forall s \in \mathcal{S}, a \in \mathcal{A}, h \in [H], k \in [K]$. Finally, $\text{Median}(\cdot)$ denotes the median of a set of numbers.

**Reinforcement learning framework** We consider episodic tabular Markov Decision Processes (MDPs) defined by a tuple $\{\mathcal{S}, \mathcal{A}, H, \{P_h\}_{h=1}^H, \{r_h\}_{h=1}^H, \nu\}$, where $\mathcal{S}$ and $\mathcal{A}$ represent the state and action spaces, respectively, with $S$ and $A$ denoting the respective cardinality of these sets. $H$ is the horizon which defines the number of steps the agent can take before the end of an episode. $P_h(\cdot|s, a) \in \Delta\mathcal{S}$ is the transition function which represents the probability of transitioning to the next state if the agent takes action $a$ at state $s$. And $r_h(s, a)$ is the reward function, accounting for the reward the agent can collect by taking action $a$ at state $s$. In this work, we assume that the reward is deterministic and bounded $r_h(s, a) \in [0, 1]$, which is consistent with prior work (Dann et al., 2023). We denote the initial state distribution by $\nu \in \Delta\mathcal{S}$.

A policy $\pi = \{\pi_h\}_{h=1}^H$ is a mapping from the state space to the probability distribution space over the action space. $\pi_h(a|s)$ denotes the probability of taking action $a$ at state $s$ and step $h$. The value function $V_h^\pi(s)$ of a policy $\pi$ is the expected total reward the agent can receive by starting from step $h$, state $s$, and following the policy $\pi$, i.e., $V_h^\pi(s) = \mathbb{E}_\pi[\sum_{l=h}^H r_l|s]$. The performance $J(\pi)$ of a policy $\pi$ is defined as the expected total reward the agent can obtain.

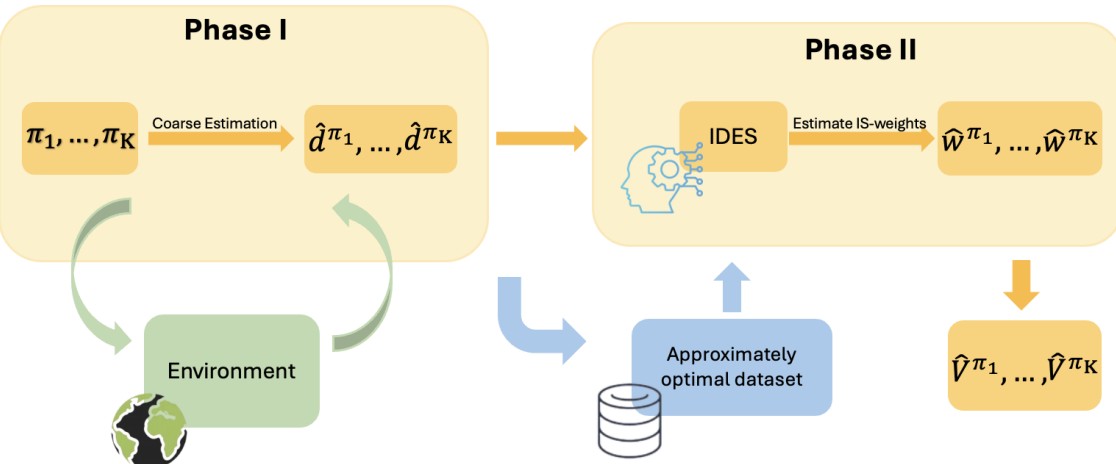

Figure 1. The scheme of CAESAR . In Phase I, we collect $\tilde{O}(1/\epsilon)$ trajectories for each target policies $\pi_1, \ldots, \pi_K$ and obtain coarse estimators of their visitation distributions $\hat{d}^{\pi_1}, \ldots, \hat{d}^{\pi_K}$. Based on the coarse estimator, we can generate an approximately optimal sampling dataset which has good coverage over the visitations of target policies. In Phase II, we sample data from the approximately optimal dataset and leverage the coarse estimators from Phase I to perform importance density estimation for each target policies by implementing IDES . With the estimated importance density $\hat{w}^{\pi_1}, \ldots, \hat{w}^{\pi_K}$, we can output the final performance evaluators $\hat{V}^{\pi_1}, \ldots, \hat{V}^{\pi_K}$.

By the definition of the value function, we have $J(\pi) = V_1^\pi(s|s \sim \nu)$. For simplicity, in the following context, we use $V_1^\pi$ to denote $V_1^\pi(s|s \sim \nu)$.

The state visitation distribution $d_h^\pi(s)$ of a policy $\pi$ represents the probability of reaching state $s$ at step $h$ if the agent starts from a state sampled from the initial state distribution $\nu$ at step $l = 1$ and follows policy $\pi$ subsequently, i.e., $d_h^\pi(s) = \mathbb{P}[s_h = s | s_1 \sim \nu, \pi]$. Similarly, the state-action visitation distribution $d_h^\pi(s, a)$ is defined as $d_h^\pi(s, a) = d_h^\pi(s)\pi(a|s)$. Based on the definition of the visitation distribution, the performance of policy $\pi$ can also be expressed as $J(\pi) = V_1^\pi = \sum_{h=1}^H \sum_{s,a} d_h^\pi(s,a)r_h(s,a)$.

**Multiple-policy evaluation problem setup** In multiple-policy evaluation, we are given a set of known policies $\{\pi^k\}_{k=1}^K$ and a pair of factors $\{\epsilon, \delta\}$. The objective is to evaluate the performance of these given policies such that with probability at least $1 - \delta$, $\forall \pi \in \{\pi^k\}_{k=1}^K$, $|\hat{V}_1^\pi - V_1^\pi| \leq \epsilon$, where $\hat{V}_1^\pi$ is the performance estimator.

## 4. Main Results and Algorithm

In this section, we introduce our main algorithm, CAESAR step-by-step and present the main results. CAESAR is sketched out in Algorithm 2 as well as in Figure 1 for easy reference. We try to build a single sampling dataset with good coverage over target policies, with which we can estimate the performance of all target policies simultaneously using importance weighting. To that end, we first

coarsely estimate the visitation distributions of target policies at the cost of a lower-order sample complexity. Based on these coarse distribution estimators, we can build an approximately optimal sampling distribution by solving a convex optimization problem. Finally, we utilize the idea of DualDICE (Nachum et al., 2019) with some modifications to estimate the importance-weighting ratio.

### 4.1. Coarse estimation of visitation distributions

It is well known that to estimate a quantity up to $\epsilon$-accuracy, approximately $\tilde{O}(\frac{1}{\epsilon^2})$ samples are needed, e.g., as indicated by Hoeffding's inequality. However, it does not serve our need since it is too expensive in terms of sample complexity. In other words, if we can estimate the visitations of target policies with $\epsilon$-accuracy, the value functions derived from these estimated visitations will already be sufficiently accurate. At the same time, designing a sampling distribution that ensures good coverage of the target policy set requires certain knowledge about the visitations of the target policies. To that end, we introduce the concept of coarse estimators which is defined below.

**Definition 4.1** (Coarse Estimator)**.** Given an accuracy $\epsilon$, An estimator $\hat{x}$ is a coarse estimator to the true value $x$ if it satisfies $|\hat{x} - x| \leq \max\{\epsilon, |x|/c\}$ where $c$ is a constant.

We next show that coarse estimation of the visitation distributions can be achieved by paying sample complexity of just $\tilde{O}(\frac{1}{\epsilon})$ which is much cheaper compared to $\tilde{O}(\frac{1}{\epsilon^2})$. And we will show in the next section that the coarse estimator pro-

vides us enough information to construct an approximately optimal sampling distribution that minimizes the maximum visitation ratio among all policies to estimate.

The idea behind the feasibility of computing these coarse estimators is based on the following lemma which shows that estimating the mean value of a Bernoulli random variable up to constant multiplicative accuracy only requires $\tilde{O}(\frac{1}{\epsilon})$ samples.

**Lemma 4.2.** *Let $Z_\ell$ be i.i.d. samples $Z_\ell \overset{i.i.d.}{\sim} \text{Ber}(p)$. Setting $t \geq \frac{C \log(C/\epsilon\delta)}{\epsilon}$, for some known constant $C > 0$, it follows that with probability at least $1 - \delta$, the empirical mean estimator $\hat{p}_t = \frac{1}{t} \sum_{\ell=1}^{t} Z_\ell$ satisfies, $|\hat{p}_t - p| \leq \max\{\epsilon, \frac{p}{4}\}$.*

Let $\{(s_1^i, a_1^i), (s_2^i, a_2^i), \dots, (s_H^i, a_H^i)\}$ denote a trajectory collected by following policy $\pi$. Then the random variable $Z_i(h, s, a)$ defined below is a Bernoulli random variable such that $Z_i(h, s, a) \overset{i.i.d.}{\sim} \text{Ber}(d_h^\pi(s, a))$.

$$Z_i(h, s, a) = \begin{cases} 1, & s_h^i, a_h^i = s, a, \\ 0, & \text{otherwise.} \end{cases}$$

We can construct such random variables for each $(h, s, a)$. Together with Lemma 4.2, we are able to coarsely estimate the visitation distributions of a policy by sampling $\tilde{O}(\frac{1}{\epsilon})$ trajectories.

**Proposition 4.3.** *With number of trajectories $n \geq \frac{CK \log(CK/\epsilon\delta)}{\epsilon} = \tilde{O}(\frac{1}{\epsilon})$, we can estimate $\hat{d}^{\pi^k} = \{\hat{d}_h^{\pi^k}\}_{h=1}^H$ such that with probability at least $1 - \delta$, $|\hat{d}_h^{\pi^k}(s, a) - d_h^{\pi^k}(s, a)| \leq \max\{\epsilon, \frac{d_h^{\pi^k}(s,a)}{4}\}$, $\forall s, a, h, k$.*

In Appendix B, we propose an algorithm, MARCH (*Multi-policy Approximation via Ratio-based Coarse Handling*), which computes coarse estimators of visitation distributions for all deterministic policies with sample complexity $\tilde{O}(\frac{\text{poly}(H,S,A)}{\epsilon})$. This is a significant result since the number of deterministic policies scales exponentially with the size of the state space and horizon. The result is achieved through a novel analysis based on our proposed notion named $\beta$−distance. Due to space constraints, we refer readers to Appendix B for further details.

Before proceeding to the next section, notice that states and actions with low estimated visitation probabilities can be ignored without inducing significant errors based on the following lemma.

**Lemma 4.4.** *Suppose we have an estimator $\hat{d}(s, a)$ of $d(s, a)$ such that $|\hat{d}(s, a) - d(s, a)| \leq \max\{\epsilon', \frac{d(s,a)}{4}\}$. If $\hat{d}(s, a) \geq 5\epsilon'$, then $\max\{\epsilon', \frac{d(s,a)}{4}\} = \frac{d(s,a)}{4}$, and if $\hat{d}(s, a) \leq 5\epsilon'$, then $d(s, a) \leq 7\epsilon'$.*

Lemma 4.4 tells us if we replace $\epsilon'$ by $\frac{\epsilon}{14SA}$, the error induced by ignoring those state-action pairs which satisfy

$\hat{d}(s, a) \leq 5\epsilon'$ is at most $\frac{\epsilon}{2}$. For simplicity of presentation, we can set $\hat{d}_h^\pi(s, a) = d_h^\pi(s, a) = 0$ if $\hat{d}_h^\pi(s, a) < \frac{5\epsilon}{14SA}$. Finally, we have coarse estimators of the visitations for target policies such that,

$$|\hat{d}_h^{\pi^k}(s, a) - d_h^{\pi^k}(s, a)| \leq \frac{d_h^{\pi^k}(s, a)}{4}, \ \forall s, a, h, k. \quad (1)$$

## 4.2. Approximately optimal sampling distribution

In this section, we validate the claim from the previous section that the coarse estimator provides sufficient information (i.e., accurate enough) to construct an approximately optimal sampling distribution that minimizes the maximum visitation ratio among all policies to estimate.

Suppose that we have a dataset with distribution $\{\mu_h\}_{h=1}^H$, from which we can sample trajectories $\{s_1^i, a_1^i, s_2^i, a_2^i, \dots, s_H^i, a_H^i\}_{i=1}^n$, then we can evaluate the expected total rewards of target policies by importance weighting. Specifically,

$$\hat{V}_1^{\pi^k} = \frac{1}{n} \sum_{i=1}^n X_i^{\pi^k}, \ k \in [K]. \quad (2)$$

where $X_i^{\pi^k} = \sum_{h=1}^H \frac{d_h^{\pi^k}(s_h^i, a_h^i)}{\mu_h(s_h^i, a_h^i)} r_h(s_h^i, a_h^i)$ is the total reward gained in a trajectory. Note that the above estimators rely on an unknown quantity $d_h^\pi(s, a)$, i.e., the true visitation distribution. In the next section, we will show how to accurately estimate these importance-weighting ratios.

It can be shown that the variance of the value function estimator $X_i^{\pi^k}$ is bounded by (see Appendix A.2),

$$\mathbb{V}_\mu[X_i^{\pi^k}] \leq H \cdot \sum_{h=1}^H \mathbb{E}_{d_h^{\pi^k}} \left[ \frac{d_h^{\pi^k}(s_h, a_h)}{\mu_h(s_h, a_h)} \right]. \quad (3)$$

We aim to identify the optimal sampling distribution by minimizing the variance (3) of the value function estimator across all target policies, resulting in the following convex optimization problem:

$$\mu_h^* = \underset{\mu_h \in \text{conv}(\mathcal{D}_h)}{\arg\min} \max_{k \in [K]} \sum_{s,a} \frac{(d_h^{\pi^k}(s, a))^2}{\mu_h(s, a)}, \ h \in [H]. \quad (4)$$

We constrain $\mu_h$ in the convex hull of $\mathcal{D}_h = \{d_h^{\pi^k} : k \in [K]\}$, since in some cases, the optimal $\mu^*$ may not be realized by any policy (see Appendix A.3). One can also set $\mathcal{D}_h = \{d_h^\pi : \pi \in \Pi_{det}\}$ which allows any feasible distribution, ensuring a globally optimal sampling distribution at the cost of more computations towards solving the optimization problem. We denote the optimal solution to (4) as $\mu_h^* = \sum_{k=1}^K \alpha_k^* d_h^{\pi^k}$.

It is impossible to solve (4) since the true visitations $d_h^\pi$ are unknown to us. Therefore, we utilize the coarse estimators obtained in the last section to replace these unknown

distributions which leads to the following approximate optimization problem,

$$\hat{\mu}_h^* = \underset{\mu_h \in \text{conv}(\hat{\mathcal{D}}_h)}{\arg\min} \max_{k \in [K]} \sum_{s,a} \frac{(\hat{d}_h^{\pi^k}(s,a))^2}{\mu_h(s,a)}, \; h \in [H], \quad (5)$$

where $\hat{\mathcal{D}}_h = \{\hat{d}_h^{\pi^k} : k \in [K]\}$. We denote the optimal solution to (5) by $\hat{\mu}_h^* = \sum_{k=1}^{K} \hat{\alpha}_K^* \hat{d}_h^{\pi^K}$. Consequently, our true sampling distribution from which trajectories are sampled is $\tilde{\mu}_h^* = \sum_{k=1}^{K} \hat{\alpha}_k^* d_h^{\pi^k}$. And based on (1), we also have the relationship,

$$|\tilde{\mu}_h^*(s,a) - \hat{\mu}_h^*(s,a)| \le \frac{\tilde{\mu}_h^*(s,a)}{4} \quad (6)$$

Finally, we conclude this section by showing $\tilde{\mu}_h^*$ is approximately optimal.

**Lemma 4.5.** *The sampling distribution $\tilde{\mu}^*$ is approximately optimal such that,*

$$\max_{k \in [K]} \sum_{s,a} \frac{(d_h^{\pi^k}(s,a))^2}{\tilde{\mu}_h^*(s,a)} \le C \max_{k \in [K]} \sum_{s,a} \frac{(d_h^{\pi^k}(s,a))^2}{\mu_h^*(s,a)}$$

*where $C$ is a constant and $\mu^*$ is the optimal solution to (4).*

### 4.3. Estimation of importance-weighting ratios

In the previous section, we constructed a sampling dataset with distribution $\tilde{\mu}^*$, from which we can draw trajectories. The remaining challenge is that estimating the value function using (2) requires knowledge of the importance-weighting ratios. To address this, we introduce an algorithm, IDES, outlined in Algorithm 1, for estimating these ratios. IDES is inspired by the idea of DualDICE (Nachum et al., 2019). In DualDICE, they propose the following loss function,

$$\ell^{\pi}(w) = \frac{1}{2}\mathbb{E}_{s,a \sim \mu}\left[w^2(s,a)\right] - \mathbb{E}_{s,a \sim d^{\pi}}\left[w(s,a)\right]. \quad (7)$$

The minimum of $\ell^{\pi}(\cdot)$ is achieved at $w^{\pi,*}(s,a) = \frac{d^{\pi}(s,a)}{\mu(s,a)}$, the importance weighting ratio. By applying a variable transformation based on Bellman's equation, they derive a final loss function that eliminates the need for on-policy samples in infinite-horizon MDPs. The importance-weighting ratios are then obtained by optimizing this loss function.

We extend this approach to finite-horizon MDPs by proposing a step-wise loss function, which we define below. It is important to emphasize that IDES is not a trivial extension of DualDICE. While IDES adopts the quadratic loss function from DualDICE, it applies fundamentally different techniques and analyses.

First, IDES leverages coarse distribution estimators to address the on-policy limitation of the second term in (7),

---

**Algorithm 1** **I**mportance **D**ensity **Es**timation (IDES )

**Input:** Horizon $H$, accuracy $\epsilon$, target policy $\pi$, coarse estimator $\{\hat{d}_h^{\pi}\}_{h=1}^{H}$, $\{\hat{\mu}_h\}_{h=1}^{H}$ and dataset $\mu$

Define feasible sets $\{D_h\}_{h=1}^{H}$ where $D_h(s,a) = [0, 2\hat{d}_h^{\pi}(s,a)]$.

Initialize $w_h^0 = 0$, $h = 1, \ldots, H$, and set $\mu_0(s_0, a_0) = 1, P_0(s|s_0, a_0) = \nu(s), \hat{w}_0 = \hat{\mu}_0 = 1$.

**for** $h = 1$ **to** $H$ **do**

  Set the iteration number of optimization, $n_h = C_h\left(\frac{H^4}{\epsilon^2} \sum_{s,a} \frac{(\hat{d}_h^{\pi}(s,a))^2}{\hat{\mu}_h(s,a)} + \frac{(\hat{d}_{h-1}^{\pi}(s,a))^2}{\hat{\mu}_{h-1}(s,a)}\right)$, where $C_h$ is a known constant.

  **for** $i = 1$ **to** $n_h$ **do**

    Sample $\{s_h^i, a_h^i\}$ from $\mu_h$ and $\{s_{h-1}^i, a_{h-1}^i, s_h^{i'}\}$ from $\mu_{h-1}$.

    Calculate gradient $g(w_h^{i-1})$,

$$g(w_h^{i-1})(s,a) = \frac{w_h^{i-1}(s,a)}{\hat{\mu}_h(s,a)}\mathbb{I}(s_h^i = s, a_h^i = a)$$
$$- \frac{\hat{w}_{h-1}(s_{h-1}^i, a_{h-1}^i)}{\hat{\mu}_{h-1}(s_{h-1}^i, a_{h-1}^i)}\pi(a|s)\mathbb{I}(s_h^{i'} = s).$$

    Update $w_h^i = Proj_{w \in D_h}\{w_h^{i-1} - \eta_h^i g(w_h^{i-1})\}$.

  **end for**

  Output the estimator $\hat{w}_h = \frac{1}{\sum_{i=1}^{n_h} i}\sum_{i=1}^{n_h} w_h^i$.

**end for**

---

whereas DualDICE relies on a variable transformation based on Bellman's equation, which is only applicable to infinite-horizon MDPs. Second, IDES employs a step-wise objective function, requiring sequential step-to-step optimization and analysis, whereas DualDICE formulates a single loss function. Third, although both IDES and DualDICE achieve a sample complexity of $\tilde{O}(C/\epsilon^2)$, the value of $C$ in DualDICE's bound is not sufficiently tight for our purposes, which involve deriving instance-dependent guarantees. In contrast, we offer a precise analysis linking the value of $C$ in IDES to the expected visitation ratios. Lastly, IDES provides high-probability results for visitation ratio estimation, whereas DualDICE's results hold only in expectation.

Our step-wise loss function of a policy $\pi$ at each step $h$ is defined as,

$$\ell_h^{\pi}(w) = \frac{1}{2}\mathbb{E}_{s,a \sim \tilde{\mu}_h^*}\left[\frac{w^2(s,a)}{\hat{\mu}_h^*(s,a)}\right]$$
$$- \mathbb{E}_{\substack{s',a' \sim \tilde{\mu}_{h-1}^* \\ s \sim P_{h-1}(\cdot|s',a')}}\left[\sum_a \frac{\hat{w}_{h-1}(s',a')}{\hat{\mu}_{h-1}^*(s',a')}w(s,a)\pi(a|s)\right]$$

where $\tilde{\mu}_h^* = \sum_{k=1}^{K} \hat{\alpha}_k^* d_h^{\pi^k}$ is the sampling distribution, and $\hat{\mu}_h^* = \sum_{k=1}^{K} \hat{\alpha}_k^* \hat{d}_h^{\pi^k}$ is the optimal solution to the approximate optimization problem (5) (see Appendix 4.2). And for

notation simplicity, we set $\tilde{\mu}_0^*(s_0, a_0) = 1, P_0(s|s_0, a_0) = \nu(s), \hat{w}_0 = \hat{\mu}_0^* = 1$.

Although it may appear complex at first glance, it possesses favorable properties that allow us to derive importance-weighting ratios iteratively, as formalized in the following two lemmas.

**Lemma 4.6.** *Suppose we have an estimator $\hat{w}_{h-1}$ at step $h-1$ such that,*

$$\sum_{s,a} \left| \tilde{\mu}_{h-1}^*(s, a) \frac{\hat{w}_{h-1}(s, a)}{\hat{\mu}_{h-1}^*(s, a)} - d_{h-1}^\pi(s, a) \right| \leq \epsilon,$$

*then by minimizing the loss function $\ell_h^\pi(w)$ at step $h$ to $\|\nabla \ell_h^\pi(\hat{w}_h(s, a))\|_1 \leq \epsilon$, we have,*

$$\sum_{s,a} \left| \tilde{\mu}_h^*(s, a) \frac{\hat{w}_h(s, a)}{\hat{\mu}_h^*(s, a)} - d_h^\pi(s, a) \right| \leq 2\epsilon.$$

Lemma 4.6 shows that the importance-weighting ratio estimator from the previous step enables the estimation of the ratio at the current step, introducing only an additive error. Consequently, by optimizing iteratively, we can accurately estimate the importance-weighting ratios at all steps, as formalized in the following lemma.

**Lemma 4.7.** *Implementing Algorithm 1 with $\pi^k$, we have the importance density estimator $\frac{\hat{w}_h^{\pi^k}(s,a)}{\hat{\mu}_h^*(s,a)}$ such that,*

$$\mathbb{E}\left[ \sum_{s,a} \left| \tilde{\mu}_h^*(s, a) \frac{\hat{w}_h^{\pi^k}(s, a)}{\hat{\mu}_h^*(s, a)} - d_h^{\pi^k}(s, a) \right| \right] \leq \frac{\epsilon}{4H}, \ \forall h. \tag{8}$$

The above result holds in expectation. To obtain a high-probability guarantee, we introduce a Median-of-Means (MoM) estimator (Minsker, 2023), formalized in the following lemma, along with a data-splitting technique. Together, these methods enable us to transform (8) into a high-probability result (see Appendix A.8).

**Lemma 4.8.** *Let $x \in \mathbb{R}$ and suppose we have a stochastic estimator $\hat{x}$ such that $\mathbb{E}[|\hat{x} - x|] \leq \frac{\epsilon}{4}$. Then, if we generate $N = O\left(\log(1/\delta)\right)$ i.i.d. estimators $\{\hat{x}_1, \hat{x}_2, \ldots, \hat{x}_N\}$ and choose $\hat{x}_{MoM} = Median(\hat{x}_1, \hat{x}_2, \ldots, \hat{x}_N)$, we have with probability at least $1 - \delta$,*

$$|\hat{x}_{MoM} - x| \leq \epsilon.$$

Another noteworthy property of our loss function is that it is $\gamma$-strongly convex and $\xi$-smooth where $\gamma = \min_{s,a} \frac{\tilde{\mu}_h^*(s,a)}{\hat{\mu}_h^*(s,a)}, \xi = \max_{s,a} \frac{\tilde{\mu}_h^*(s,a)}{\hat{\mu}_h^*(s,a)}$. From property (6), it follows that $\gamma$ and $\xi$ are bounded from both sides, as is their ratio $\frac{\xi}{\gamma}$. This property plays a crucial role in analyzing the sampling complexity of the minimization of the loss function via stochastic gradient descent, which we discuss in Appendix A.6.

---

**Algorithm 2** **C**oarse and **A**daptive **ES**timation with **A**pproximate **R**eweighing for Multi-Policy Evaluation (CAESAR )

---

**Input:** Accuracy $\epsilon$, confidence $\delta$, target policies $\{\pi^k\}_{k=1}^K$.

Coarsely estimate visitation distributions of all target policies and get $\{\hat{d}^{\pi^k} : k \in [K]\}$.

Solve the approximate optimization problem (5) and construct the dataset $\tilde{\mu}^*$.

Implement IDES (Algorithm 1) to get importance-weighting ratios $\{\hat{w}^{\pi^k}\}_{k=1}^K$.

Build the final performance estimator $\{\hat{V}_1^{\pi^k}\}_{k=1}^K$ by (9).

**Output:** $\{\hat{V}_1^{\pi^k}\}_{k=1}^K$.

---

### 4.4. Main results

We are now ready to present our main sample complexity result for the multiple-policy evaluation problem, building on the findings from previous sections. Using the importance-weighting ratio estimator $\frac{\hat{w}_h^{\pi^k}(s,a)}{\hat{\mu}_h^*(s,a)}$, we can evaluate the performance of the target policy $\pi^k$ by,

$$\hat{V}_1^{\pi^k} = \frac{1}{n} \sum_{i=1}^n \sum_{h=1}^H \frac{\hat{w}_h^{\pi^k}(s_h^i, a_h^i)}{\hat{\mu}_h^*(s_h^i, a_h^i)} r_h(s_h^i, a_h^i). \tag{9}$$

where $\{s_h^i, a_h^i\}_{i=1}^n$ is drawn from our approximately optimal sampling distribution $\tilde{\mu}_h^*$.

The following theorem presents our main results on the sample complexity of CAESAR. We will leave the detailed derivation to Appendix A.8.

**Theorem 4.9.** *Implementing Algorithm 2. Then, with probability at least $1 - \delta$, for all target policies, we have that $|\hat{V}_1^{\pi^k} - V_1^{\pi^k}| \leq \epsilon$. And the total number of trajectories sampled is,*

$$n = \tilde{O}\left( \frac{H^4}{\epsilon^2} \sum_{h=1}^H \max_{k \in [K]} \sum_{s,a} \frac{(d_h^{\pi^k}(s,a))^2}{\mu_h^*(s,a)} \right). \tag{10}$$

*where $\mu_h^*$ is the optimal solution to (4) (refer to Section 4.2). Furthermore, the above bound still holds if replacing the unknown true visitation distributions $\{\{d^{\pi^k}\}_{k=1}^K, \mu^*\}$ by the coarse estimators $\{\{\hat{d}^{\pi^k}\}_{k=1}^K, \hat{\mu}^*\}$ which can provide us a concrete value of the sample complexity.*

We provide an instance-dependent result that characterizes the complexity of the multiple-policy evaluation problem based on the overlap of the visitation distributions of the target policies, aligning with intuitions. In the special case where all target policies are identical, i.e., single-policy evaluation, our sample complexity is $\tilde{O}(\frac{\text{poly}(H)}{\epsilon^2})$, independent of $S$ and $A$, which is consistent with the result of Monte

Carlo estimation. Furthermore, if the target policies are deterministic, we can establish an instance-independent upper bound on our sample complexity, as formalized in the following corollary.

**Corollary 4.10.** *If the target policies to be evaluated are deterministic, the sample complexity of* CAESAR *is always bounded by* $\tilde{O}\left(\frac{poly(H)S^2A}{\epsilon^2}\right)$.

Corollary 4.10 tells us that with at most $\tilde{O}\left(\frac{\text{poly}(H)S^2A}{\epsilon^2}\right)$ trajectories, by implementing CAESAR , we can evaluate all deterministic policies up to $\epsilon$ accuracy under any reward which means we can identify an $\epsilon-$ optimal policy for any reward. This is consistent with the result in Jin et al. (2020), which proposes a reward-free exploration algorithm with the same sample complexity of $\tilde{O}\left(\frac{\text{poly}(H)S^2A}{\epsilon^2}\right)$.

## 5. Discussions

In this section, we first compare our results with the existing work (Dann et al., 2023). Additionally, we explore the application of CAESAR to policy identification beyond policy evaluation.

### 5.1. Comparison with existing result

Dann et al. (2023) proposes an algorithm for multiple-policy evaluation based on the idea of trajectory stitching and achieved an instance-dependent sample complexity,

$$\tilde{O}\left(\frac{H^2}{\epsilon^2}\mathbb{E}\left[\sum_{(s,a)\in\mathcal{K}^{1:H}}\frac{1}{d^{max}(s)}\right]\right), \qquad (11)$$

where $d^{max}(s) = \max_{k\in[K]} d^{\pi^k}(s)$ and $\mathcal{K}^h \subseteq \mathcal{S} \times \mathcal{A}$ keeps track of which state-action pairs at step $h$ are visited by target policies in their trajectories.

A significant issue with the result by Dann et al. (2023) is the presence of the unfavorable $\frac{1}{d^{max}(s)}$, which can induce an undesirable dependency on $K$.

To illustrate this, consider an example of an MDP with two layers: a single initial state $s_{1,1}$ in the first layer and two terminal states in the second layer $s_{2,1}, s_{2,2}$. The transition function is the same for all actions, i.e., $P(s_{2,1}|s_{1,1},a) = p$ and $p$ is sufficiently small. Agents only receive rewards at state $s_{2,1}$, regardless of the actions they take. Hence, to evaluate the performance of a policy under this MDP, it is sufficient to consider only the second layer. Now, suppose we have $K$ target policies to evaluate, where each policy takes different actions at state $s_{1,1}$ but the same action at any state in the second layer. Since the transition function at state $s_{1,1}$ is the same for any action, the visitation distribution at state $s_{2,1}$ of all target policies is identical. Given that

$p$ is sufficiently small, the probability of reaching $s_{2,1}$ is $\mathbb{P}[s_{2,1} \in \mathcal{K}^2] = 1 - (1-p)^K \approx pK$.

According to the result (11) by Dann et al. (2023), the sample complexity in this scenario is $\tilde{O}(\frac{K}{\epsilon^2})$ which depends on $K$. In contrast, since the visitation distribution at the second layer of all target policies is identical, our result provides a sample complexity of $\tilde{O}(\frac{1}{\epsilon^2})$ without dependency on $K$.

Beyond sample complexity, our work tackles the problem from a different perspective, which complements the existing results. Our algorithm first constructs an approximately optimal dataset and then uses it to perform offline evaluation. In other words, we extend the offline evaluation framework to multiple-policy setting. In contrast, (Dann et al., 2023) evaluates policies in an online and on-policy manner.

### 5.2. Near-optimal policy identification

Besides policy evaluation, CAESAR can also be applied to identify a near-optimal policy. Fixing the high-probability factor, we denote the sample complexity of CAESAR by $\tilde{O}(\frac{\Theta(\Pi)}{\gamma^2})$, where $\Pi$ is the set of policies to be evaluated and $\gamma$ is the estimation error. We provide a simple algorithm based on CAESAR in Appendix D that achieves an instance-dependent sample complexity $\tilde{O}(\max_{\gamma\geq\epsilon}\frac{\Theta(\Pi_\gamma)}{\gamma^2})$ to identify a $\epsilon-$optimal policy, where $\Pi_\gamma = \{\pi : V_1^* - V_1^\pi \leq 8\gamma\}$. This result is interesting as it offers a different perspective beyond the existing gap-dependent results (Simchowitz and Jamieson, 2019; Dann et al., 2021). Furthermore, this result can be easily extended to the multi-reward setting. Due to space constraints, we leave the detailed discussion to Appendix D.

## 6. Conclusion and Future Work

In this work, we consider the problem of multi-policy evaluation. We propose an algorithm, CAESAR, based on computing an approximately optimal sampling dataset and using the data sampled from it to perform the simultaneous estimation of the policy values. The algorithm consists of three techniques. First, we obtain coarse distribution estimators at a lower-order sample cost. Second, based on the coarse estimator, we obtain an approximately optimal sampling dataset. Lastly, we propose a step-wise loss function to estimate the importance weighting ratios.

Beyond the results of this work, there are still some open questions of interest. First, our sample complexity has a dependency on $H^4$ which is induced by the error propagation in the estimation of the importance weighting ratios. We conjecture a dependency on $H^2$ is possible by considering a comprehensive loss function instead of step-wise loss functions. Second, considering a reward-dependent

and variance-aware sample complexity is also an interesting direction. Third, it is still a challenging problem to derive the lower bound for multiple-policy evaluation. Finally, we are interested to see what other uses the research community may find for coarse distribution estimation.

## Acknowledgements

This work was supported in part by funding from the Eric and Wendy Schmidt Center at the Broad Institute of MIT and Harvard, the NSF under grants CCF-2200052 and IIS-1914792, the ONR under grants N00014-19-1-2571 and N00014-21-1-2844, the NIH under grant 1UL1TR001430, the DOE under grant DE-AC02-05CH11231, the ARPA-E under grant and DE-AR0001282, and the Boston University Kilachand Fund for Integrated Life Science and Engineering.

## Impact Statement

This work aims to advance the field of Machine Learning by providing new insights and methodologies for multiple-policy evaluation in reinforcement learning. Our contributions enhance the theoretical understanding of policy evaluation. While our research has broad implications, we do not identify any specific societal consequences that require particular emphasis at this time.

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

# A. Proof of theorems and lemmas in Section 4

## A.1. Proof of Lemma 4.2

Our results relies on the following variant of Bernstein inequality for martingales, or Freedman's inequality (Freedman, 1975), as stated in e.g., (Agarwal et al., 2014; Beygelzimer et al., 2011).

**Lemma A.1** (Simplified Freedman's inequality). *Let $X_1, ..., X_T$ be a bounded martingale difference sequence with $|X_\ell| \leq R$. For any $\delta' \in (0,1)$, and $\eta \in (0, 1/R)$, with probability at least $1 - \delta'$,*

$$\sum_{\ell=1}^{T} X_\ell \leq \eta \sum_{\ell=1}^{T} \mathbb{E}_\ell[X_\ell^2] + \frac{\log(1/\delta')}{\eta}. \tag{12}$$

*where $\mathbb{E}_\ell[\cdot]$ is the conditional expectation[2] induced by conditioning on $X_1, \cdots, X_{\ell-1}$.*

**Lemma A.2** (Anytime Freedman). *Let $\{X_t\}_{t=1}^{\infty}$ be a bounded martingale difference sequence with $|X_t| \leq R$ for all $t \in \mathbb{N}$. For any $\delta' \in (0,1)$, and $\eta \in (0, 1/R)$, there exists a universal constant $C > 0$ such that for all $t \in \mathbb{N}$ simultaneously with probability at least $1 - \delta'$,*

$$\sum_{\ell=1}^{t} X_\ell \leq \eta \sum_{\ell=1}^{t} \mathbb{E}_\ell[X_\ell^2] + \frac{C \log(t/\delta')}{\eta}. \tag{13}$$

*where $\mathbb{E}_\ell[\cdot]$ is the conditional expectation induced by conditioning on $X_1, \cdots, X_{\ell-1}$.*

*Proof.* This result follows from Lemma A.1. Fix a time-index $t$ and define $\delta_t = \frac{\delta'}{12t^2}$. Lemma A.1 implies that with probability at least $1 - \delta_t$,

$$\sum_{\ell=1}^{t} X_\ell \leq \eta \sum_{\ell=1}^{t} \mathbb{E}_\ell\left[X_\ell^2\right] + \frac{\log(1/\delta_t)}{\eta}.$$

A union bound implies that with probability at least $1 - \sum_{\ell=1}^{t} \delta_t \geq 1 - \delta'$,

$$\sum_{\ell=1}^{t} X_\ell \leq \eta \sum_{\ell=1}^{t} \mathbb{E}_\ell\left[X_\ell^2\right] + \frac{\log(12t^2/\delta')}{\eta}$$

$$\overset{(i)}{\leq} \eta \sum_{\ell=1}^{t} \mathbb{E}_\ell\left[X_\ell^2\right] + \frac{C \log(t/\delta')}{\eta}.$$

holds for all $t \in \mathbb{N}$. Inequality $(i)$ holds because $\log(12t^2/\delta') = \mathcal{O}\left(\log(t\delta')\right)$.

$\square$

**Proposition A.3.** *Let $\delta' \in (0,1)$, $\beta \in (0,1]$ and $Z_1, \cdots, Z_T$ be an adapted sequence satisfying $0 \leq Z_\ell \leq \tilde{B}$ for all $\ell \in \mathbb{N}$. There is a universal constant $C' > 0$ such that,*

$$(1 - \beta) \sum_{t=1}^{T} \mathbb{E}_t[Z_t] - \frac{2\tilde{B}C' \log(T/\delta')}{\beta} \leq \sum_{\ell=1}^{T} Z_\ell \leq (1 + \beta) \sum_{t=1}^{T} \mathbb{E}_t[Z_t] + \frac{2\tilde{B}C' \log(T/\delta')}{\beta}$$

*with probability at least $1 - 2\delta'$ simultaneously for all $T \in \mathbb{N}$.*

*Proof.* Consider the martingale difference sequence $X_t = Z_t - \mathbb{E}_t[Z_t]$. Notice that $|X_t| \leq \tilde{B}$. Using the inequality of

---

[2]We will use this notation to denote conditional expectations throughout this work.

Lemma A.2 we obtain that for all $\eta \in (0, 1/B^2)$

$$\sum_{\ell=1}^{t} X_\ell \leq \eta \sum_{\ell=1}^{t} \mathbb{E}_\ell[X_\ell^2] + \frac{C \log(t/\delta')}{\eta}$$

$$\overset{(i)}{\leq} 2\eta B^2 \sum_{\ell=1}^{t} \mathbb{E}_\ell[Z_\ell] + \frac{C \log(t/\delta')}{\eta},$$

for all $t \in \mathbb{N}$ with probability at least $1 - \delta'$. Inequality $(i)$ holds because $\mathbb{E}_t[X_t^2] \leq B^2 \mathbb{E}[|X_t|] \leq 2B^2 \mathbb{E}_t[Z_t]$ for all $t \in \mathbb{N}$. Setting $\eta = \frac{\beta}{2B^2}$ and substituting $\sum_{\ell=1}^{t} X_\ell = \sum_{\ell=1}^{t} Z_\ell - \mathbb{E}_\ell[Z_\ell]$,

$$\sum_{\ell=1}^{t} Z_\ell \leq (1+\beta) \sum_{\ell=1}^{t} \mathbb{E}_\ell[Z_\ell] + \frac{2B^2 C \log(t/\delta')}{\beta} \tag{14}$$

with probability at least $1 - \delta'$. Now consider the martingale difference sequence $X_t' = \mathbb{E}[Z_t] - Z_t$ and notice that $|X_t'| \leq B^2$. Using the inequality of Lemma A.2 we obtain for all $\eta \in (0, 1/B^2)$,

$$\sum_{\ell=1}^{t} X_\ell' \leq \eta \sum_{\ell=1}^{t} \mathbb{E}_\ell[(X_\ell')^2] + \frac{C \log(t/\delta')}{\eta}$$

$$\leq 2\eta B^2 \sum_{\ell=1}^{t} \mathbb{E}_\ell[Z_\ell] + \frac{C \log(t/\delta')}{\eta}.$$

Setting $\eta = \frac{\beta}{2B^2}$ and substituting $\sum_{\ell=1}^{t} X_\ell' = \sum_{\ell=1}^{t} \mathbb{E}[Z_\ell] - Z_\ell$ we have,

$$(1-\beta) \sum_{\ell=1}^{t} \mathbb{E}[Z_\ell] \leq \sum_{\ell=1}^{t} Z_\ell + \frac{2B^2 C \log(t/\delta')}{\beta} \tag{15}$$

with probability at least $1 - \delta'$. Combining Equations 14 and 15 and using a union bound yields the desired result.

$\square$

Let the $Z_\ell$ be i.i.d. samples $Z_\ell \overset{i.i.d.}{\sim} \text{Ber}(p)$. The empirical mean estimator, $\widehat{p}_t = \frac{1}{t} \sum_{\ell=1}^{t} Z_\ell$ satisfies,

$$(1-\beta)p - \frac{2C' \log(t/\delta')}{\beta t} \leq \widehat{p}_t \leq (1+\beta)p + \frac{2C' \log(t/\delta')}{\beta t}$$

with probability at least $1 - 2\delta'$ for all $t \in \mathbb{N}$ where $C' > 0$ is a (known) universal constant. Given $\epsilon > 0$ set $t \geq \frac{8C' \log(t/\delta')}{\beta \epsilon}$ (notice the dependence of $t$ on the RHS - this can be achieved by setting $t \geq \frac{C \log(C/\beta \epsilon \delta')}{\beta \epsilon}$ for some (known) universal constant $C > 0$).

In this case observe that,

$$(1-\beta)p - \epsilon/8 \leq \widehat{p}_t \leq (1+\beta)p + \epsilon/8.$$

Setting $\beta = 1/8$,

$$7p/8 - \epsilon/8 \leq \widehat{p}_t \leq 9p/8 + \epsilon/8,$$

so that,

$$p - \widehat{p}_t \leq p/8 + \epsilon/8,$$

and

$$\widehat{p}_t - p \leq p/8 + \epsilon/8,$$

which implies $|\widehat{p}_t - p| \leq p/8 + \epsilon/8 \leq 2 \max(p/8, \epsilon/8) = \max(p/4, \epsilon/4)$.

## A.2. Derivation of the optimal sampling distribution

Our performance estimator is,

$$\hat{V}_1^{\pi^k} = \frac{1}{n} \sum_{i=1}^{n} \sum_{h=1}^{H} \frac{d_h^{\pi^k}(s_h^i, a_h^i)}{\mu_h(s_h^i, a_h^i)} r(s_h^i, a_h^i), \ k \in [K].$$

Denote $\sum_{h=1}^{H} \frac{d_h^{\pi^k}(s_h^i, a_h^i)}{\mu_h(s_h^i, a_h^i)} r_h(s_h^i, a_h^i)$ by $X_i$. And for simplicity, denote $\mathbb{E}_{(s_1,a_1)\sim\mu_1,\ldots,(s_H,a_H)\sim\mu_H}$ by $\mathbb{E}_\mu$, the variance of our estimator is bounded by,

$$\mathbb{E}_\mu[X_i^2] = \mathbb{E}_\mu \left[ \left( \sum_{h=1}^{H} \frac{d_h^{\pi^k}(s_h^i, a_h^i)}{\mu_h(s_h^i, a_h^i)} r_h(s_h^i, a_h^i) \right)^2 \right]$$

$$\leq \mathbb{E}_\mu \left[ H \cdot \sum_{h=1}^{H} \left( \frac{d_h^{\pi^k}(s_h^i, a_h^i)}{\mu_h(s_h^i, a_h^i)} r_h(s_h^i, a_h^i) \right)^2 \right]$$

$$\leq \mathbb{E}_\mu \left[ H \cdot \sum_{h=1}^{H} \left( \frac{d_h^{\pi^k}(s_h^i, a_h^i)}{\mu_h(s_h^i, a_h^i)} \right)^2 \right]$$

$$= H \cdot \sum_{h=1}^{H} \mathbb{E}_{d_h^{\pi^k}} \left[ \frac{d_h^{\pi^k}(s_h^i, a_h^i)}{\mu_h(s_h^i, a_h^i)} \right].$$

The first inequality holds by $\mathrm{Cauchy} - \mathrm{Schwarz}$ inequality. The second inequality holds due to the assumption $r_h(s, a) \in [0, 1]$.

Denote $\sum_{h=1}^{H} \mathbb{E}_{d_h^{\pi^k}} \left[ \frac{d_h^{\pi^k}(s_h^i, a_h^i)}{\mu_h(s_h^i, a_h^i)} \right]$ by $\rho_{\mu,k}$. Applying $\mathrm{Bernstein}$'s inequality, we have that with probability at least $1 - \delta$ and $n$ samples, it holds,

$$|\hat{V}_1^{\pi^k} - V_1^{\pi^k}| \leq \sqrt{\frac{2H\rho_{\mu,k}\log(1/\delta)}{n}} + \frac{2M_k \log(1/\delta)}{3n},$$

where $M_k = \max_{s_1,a_1,\ldots,s_H,a_H} \sum_{h=1}^{H} \frac{d_h^{\pi^k}(s_h,a_h)}{\mu_h(s_h,a_h)} r_h(s_h, a_h)$.

To achieve an $\epsilon$ accuracy of evaluation, we need samples,

$$n_{\mu,k} \leq \frac{8H\rho_{\mu,k}\log(1/\delta)}{\epsilon^2} + \frac{4M_k \log(1/\delta)}{3\epsilon}.$$

Take the union bound over all target policies,

$$n_\mu \leq \frac{8H \max_{k\in[K]} \rho_{\mu,k} \log(K/\delta)}{\epsilon^2} + \frac{4M \log(K/\delta)}{3\epsilon},$$

where $M = \max_{k\in[K]} M_k$.

We define the optimal sampling distribution $\mu^*$ as the one minimizing the higher order sample complexity,

$$\mu_h^* = \arg\min_{\mu_h} \max_{k\in[K]} \mathbb{E}_{d_h^{\pi^k}(s,a)} \left[ \frac{d_h^{\pi^k}(s, a)}{\mu_h(s, a)} \right]$$

$$= \arg\min_{\mu_h} \max_{k\in[K]} \sum_{s,a} \frac{\left( d_h^{\pi^k}(s, a) \right)^2}{\mu_h(s, a)}, \ h = 1, \ldots, H.$$

### A.3. An example of unrealizable optimal sampling distribution

Here, we give an example to illustrate the assertation that in some cases, the optimal sampling distribution cannot be realized by any policy.

Consider such a MDP with two layers, in the first layer, there is a single initial state $s_{1,1}$, in the second layer, there are two states $s_{2,1}, s_{2,2}$. The transition function at state $s_{1,1}$ is identical for any action, $\mathbb{P}(s_{2,1}|s_{1,1}, a) = \mathbb{P}(s_{2,2}|s_{1,1}, a) = \frac{1}{2}$. Hence, for any policy, the only realizable state visitation distribution at the second layer is $d_2(s_{2,1}) = d_2(s_{2,2}) = \frac{1}{2}$.

Suppose the target policies take $K \geq 2$ different actions at state $s_{2,1}$ while take the same action at state $s_{2,2}$.

By solving the optimization problem, we have the optimal sampling distribution at the second layer,

$$\mu_2^*(s_{2,1}) = \frac{K^2}{1 + K^2}, \ \mu_2^*(s_{2,2}) = \frac{1}{1 + K^2},$$

which is clearly not realizable by any policy.

### A.4. Proof of Lemma 4.5

By property (1) and (6), we have $\frac{4}{5}\hat{d}_h^{\pi^k}(s, a) \leq d_h^{\pi^k}(s, a) \leq \frac{4}{3}\hat{d}_h^{\pi^k}(s, a)$ and $\frac{4}{5}\hat{\mu}_h^*(s, a) \leq \tilde{\mu}_h^*(s, a) \leq \frac{4}{3}\hat{\mu}_h^*(s, a)$. Hence,

$$\max_{k \in [K]} \sum_{s,a} \frac{(d_h^{\pi^k}(s, a))^2}{\tilde{\mu}_h^*(s, a)} \leq \frac{25}{12} \max_{k \in [K]} \sum_{s,a} \frac{(\hat{d}_h^{\pi^k}(s, a))^2}{\hat{\mu}_h^*(s, a)}$$

Remember $\mu^*$ is of the form $\sum_{k=1}^K \alpha_k^* d^{\pi^k}$. Let $\mu'$ be $\sum_{k=1}^K \alpha_k^* \hat{d}^{\pi^k}$. Then we have $|\mu' - \mu^*| \leq \max\{\epsilon, \frac{\mu^*}{4}\}$ and $\mu'$ is in the feasible set $\hat{\mathcal{D}}_h = \{\hat{d}_h^{\pi^k} : k \in [K]\}$. Since $\hat{\mu}^*$ is the optimal solution to the approximate optimization problem (5), we have,

$$\max_{k \in [K]} \sum_{s,a} \frac{(\hat{d}_h^{\pi^k}(s, a))^2}{\hat{\mu}_h^*(s, a)} \leq \max_{k \in [K]} \sum_{s,a} \frac{(\hat{d}_h^{\pi^k}(s, a))^2}{\mu_h'(s, a)} \leq \frac{25}{12} \max_{k \in [K]} \sum_{s,a} \frac{(d_h^{\pi^k}(s, a))^2}{\mu_h^*(s, a)}$$

The second inequality is again based on the property of coarse estimators. Together, we have,

$$\max_{k \in [K]} \sum_{s,a} \frac{(d_h^{\pi^k}(s, a))^2}{\tilde{\mu}_h^*(s, a)} \leq C \max_{k \in [K]} \sum_{s,a} \frac{(d_h^{\pi^k}(s, a))^2}{\mu_h^*(s, a)}.$$

### A.5. Proof of Lemma 4.6

*Proof.* The gradient of $\ell_h^\pi(w)$ is,

$$\nabla_{w(s,a)}\ell_h^\pi(w) = \frac{\tilde{\mu}_h(s, a)}{\hat{\mu}_h(s, a)}w(s, a) - \sum_{s',a'} \tilde{\mu}_{h-1}(s', a')P(s|s', a')\pi(a|s)\frac{\hat{w}_{h-1}(s', a')}{\hat{\mu}_{h-1}(s', a')}.$$

Suppose by some SGD algorithm, we can converge to a point $\hat{w}_h$ such that the gradient of the loss function is less than $\epsilon$,

$$\|\nabla\ell_h^\pi(\hat{w}_h)\|_1 = \sum_{s,a} \left| \frac{\tilde{\mu}_h(s, a)}{\hat{\mu}_h(s, a)}\hat{w}_h(s, a) - \sum_{s',a'} \tilde{\mu}_{h-1}(s', a')P(s|s', a')\pi(a|s)\frac{\hat{w}_{h-1}(s', a')}{\hat{\mu}_{h-1}(s', a')} \right| \leq \epsilon.$$

By decomposing,

$$\left| \frac{\tilde{\mu}_h(s,a)}{\hat{\mu}_h(s,a)} \hat{w}_h(s,a) - \sum_{s',a'} \tilde{\mu}_{h-1}(s',a') P(s|s',a') \pi(a|s) \frac{\hat{w}_{h-1}(s',a')}{\hat{\mu}_{h-1}(s',a')} \right|$$

$$= \left| \frac{\tilde{\mu}_h(s,a)}{\hat{\mu}_h(s,a)} \hat{w}_h(s,a) - d_h^\pi(s,a) + d_h^\pi(s,a) - \sum_{s',a'} \tilde{\mu}_{h-1}(s',a') P(s|s',a') \pi(a|s) \frac{\hat{w}_{h-1}(s',a')}{\hat{\mu}_{h-1}(s',a')} \right|$$

$$\geq \left| \frac{\tilde{\mu}_h(s,a)}{\hat{\mu}_h(s,a)} \hat{w}_h(s,a) - d_h^\pi(s,a) \right| - \left| d_h^\pi(s,a) - \sum_{s',a'} \tilde{\mu}_{h-1}(s',a') P(s|s',a') \pi(a|s) \frac{\hat{w}_{h-1}(s',a')}{\hat{\mu}_{h-1}(s',a')} \right|$$

$$= \left| \tilde{\mu}_h(s,a) \frac{\hat{w}_h(s,a)}{\hat{\mu}_h(s,a)} - d_h^\pi(s,a) \right|$$

$$- \left| \sum_{s',a'} P(s|s',a') \pi(a|s) \left( d_{h-1}^\pi(s',a') - \tilde{\mu}_{h-1}(s',a') \frac{\hat{w}_{h-1}(s',a')}{\hat{\mu}_{h-1}(s',a')} \right) \right|.$$

Hence, we have,

$$\sum_{s,a} \left| \tilde{\mu}_h(s,a) \frac{\hat{w}_h(s,a)}{\hat{\mu}_h(s,a)} - d_h^\pi(s,a) \right|$$

$$\leq \epsilon + \sum_{s,a} \left| \sum_{s',a'} P(s|s',a') \pi(a|s) \left( d_{h-1}^\pi(s',a') - \tilde{\mu}_{h-1}(s',a') \frac{\hat{w}_{h-1}(s',a')}{\hat{\mu}_{h-1}(s',a')} \right) \right|$$

$$\leq \epsilon + \sum_{s',a'} \left| d_{h-1}^\pi(s',a') - \tilde{\mu}_{h-1}(s',a') \frac{\hat{w}_{h-1}(s',a')}{\hat{\mu}_{h-1}(s',a')} \right|$$

$$\leq 2\epsilon.$$

$\square$

## A.6. Proof of Lemma 4.7

*Proof.* The minimum $w_h^*$ of the loss function $\ell_h^\pi(w)$ is $w_h^*(s,a) = \frac{d_h^\pi(s,a)}{\tilde{\mu}_h(s,a)} \hat{\mu}_h(s,a)$ if $\hat{w}_{h-1}$ achieves optimum. By the property of the coarse distribution estimator, we have,

$$w_h^*(s,a) = \frac{d_h^\pi(s,a)}{\tilde{\mu}_h(s,a)} \hat{\mu}_h(s,a) \leq \frac{\frac{4}{3}\hat{d}_h^\pi(s,a)}{\frac{4}{5}\hat{\mu}_h(s,a)} \hat{\mu}_h(s,a) = \frac{5}{3}\hat{d}_h^\pi(s,a).$$

We can define a feasible set for the optimization problem, i.e. $w_h(s,a) \in [0, D_h(s,a)]$, $D_h(s,a) = 2\hat{d}_h^\pi(s,a)$.

Next, we analyse the variance of the stochastic gradient. We denote the stochastic gradient as $g_h(w)$, $\{s_1^i, a_1^i, \ldots, s_H^i, a_H^i\}$ a trajectory sampled from $\tilde{\mu}_h$ and $\{s_1^j, a_1^j, \ldots, s_H^j, a_H^j\}$ a trajectory sampled from $\tilde{\mu}_{h-1}$.

$$g_h(w)(s,a) = \frac{w(s,a)}{\hat{\mu}_h(s,a)} \mathbb{I}(s_h^i = s, a_h^i = a) - \frac{\hat{w}_{h-1}(s_{h-1}^j, a_{h-1}^j)}{\hat{\mu}_{h-1}(s_{h-1}^j, a_{h-1}^j)} \pi(a|s) \mathbb{I}(s_h^j = s).$$

The variance bound becomes

$$\mathbb{V}[g_h(w)] \leq \mathbb{E}[\|g_h(w)\|^2] \leq \sum_{s,a} \tilde{\mu}_h(s,a) \left( \frac{w(s,a)}{\hat{\mu}_h(s,a)} \right)^2 + \tilde{\mu}_{h-1}(s,a) \left( \frac{\hat{w}_{h-1}(s,a)}{\hat{\mu}_{h-1}(s,a)} \right)^2$$

$$\leq O\left( \sum_{s,a} \frac{(\hat{d}_h^\pi(s,a))^2}{\hat{\mu}_h(s,a)} + \frac{(\hat{d}_{h-1}^\pi(s,a))^2}{\hat{\mu}_{h-1}(s,a)} \right), \tag{16}$$

where the last inequality is due to the bounded feasible set for $w$ and the property of coarse distribution estimator $\tilde{\mu}_h(s,a) \leq \frac{4}{3}\hat{\mu}_h(s,a)$.

Based on the error propagation lemma 4.6, if we can achieve $\|\nabla \ell_h^\pi(\hat{w}_h)\|_1 \leq \frac{\epsilon}{4H^2}$ from step $h = 1$ to step $h = H$, then we have,

$$\sum_{s,a} \left| \tilde{\mu}_h(s,a) \frac{\hat{w}_h(s,a)}{\hat{\mu}_h(s,a)} - d_h^\pi(s,a) \right| \leq \frac{\epsilon}{4H}, \forall h = 1, 2, \ldots, H,$$

which can enable us to build the final estimator of the performance of policy $\pi$ with at most error $\epsilon$.

By the property of smoothness, to achieve $\|\nabla \ell_h^\pi(\hat{w}_h)\|_1 \leq \frac{\epsilon}{4H^2}$, we need to achieve $\ell_h^\pi(\hat{w}_h) - \ell_h^\pi(w_h^*) \leq \frac{\epsilon^2}{32\xi H^4}$ where $\xi$ is the smoothness factor, because,

$$\|\nabla \ell_h^\pi(\hat{w}_h)\|_1^2 \leq 2\xi(\ell_h^\pi(\hat{w}_h) - \ell_h^\pi(w_h^*)) \leq \frac{\epsilon^2}{16H^4}.$$

**Lemma A.4.** *For a $\lambda-$strongly convex loss function $L(w)$ satisfying $\|w^*\| \leq D$ for some known $D$, there exists a stochastic gradient descent algorithm that can output $\hat{w}$ after $T$ iterations such that,*

$$\mathbb{E}[L(\hat{w}) - L(w^*)] \leq \frac{2G^2}{\lambda(T+1)},$$

*where $G^2$ is the variance bound of the stochastic gradient.*

Invoke the convergence rate for strongly-convex and smooth loss functions, i.e. Lemma A.4, we have that the number of samples needed to achieve $\ell_h^\pi(\hat{w}_h) - \ell_h^\pi(w_h^*) \leq \frac{\epsilon^2}{32\xi H^4}$ is,

$$n = O\left( \frac{\xi}{\gamma} \frac{H^4 G^2}{\epsilon^2} \right).$$

We have shown in Section 4.3 that $\frac{\xi}{\gamma} \leq \frac{5}{3}$, this nice property helps us to get rid of the undesired ratio of the smoothness factor and the strongly-convexity factor, i.e. $\frac{\max_{s,a} \mu(s,a)}{\min_{s,a} \mu(s,a)}$ of the original loss function (7) which can be extremely bad. Replacing $G^2$ by our variance bound (16), we have,

$$n_h^\pi = O\left( \frac{H^4}{\epsilon^2} \left( \sum_{s,a} \frac{(\hat{d}_h^\pi(s,a))^2}{\hat{\mu}_h(s,a)} + \frac{(\hat{d}_{h-1}^\pi(s,a))^2}{\hat{\mu}_{h-1}(s,a)} \right) \right).$$

For each step $h$, we need the above number of trajectories, sum over $h$, we have the total sample complexity,

$$n^\pi = O\left( \frac{H^4}{\epsilon^2} \sum_{h=1}^H \sum_{s,a} \frac{(\hat{d}_h^\pi(s,a))^2}{\hat{\mu}_h(s,a)} \right).$$

To evaluate $K$ policies, we need trajectories,

$$n = O\left( \frac{H^4}{\epsilon^2} \sum_{h=1}^H \max_{k \in [K]} \sum_{s,a} \frac{(\hat{d}_h^{\pi^k}(s,a))^2}{\hat{\mu}_h(s,a)} \right).$$

$\square$

## A.7. Proof of Lemma 4.8

*Proof.* By Markov's inequality, we have,

$$\mathbb{P}(|\hat{\mu} - \mu| \geq \epsilon) \leq \frac{\mathbb{E}[|\hat{\mu} - \mu|]}{\epsilon} \leq \frac{1}{4}.$$

The event that $|\hat{\mu}_{MoM} - \mu| > \epsilon$ belongs to the event where more than half estimators $\hat{\mu}_i$ are outside of the desired range $|\hat{\mu}_i - \mu| > \epsilon$, hence, we have,

$$\mathbb{P}(|\hat{\mu}_{MoM} - \mu| > \epsilon) \leq \mathbb{P}(\sum_{i=1}^{N} \mathbb{I}(|\hat{\mu}_i - \mu| > \epsilon) \geq \frac{N}{2}).$$

Denote $\mathbb{I}(|\hat{\mu}_i - \mu| > \epsilon)$ by $Z_i$ and $\mathbb{E}[Z_i] = p$,

$$\begin{aligned}
\mathbb{P}(|\hat{\mu}_{MoM} - \mu| > \epsilon) &= \mathbb{P}(\sum_{i=1}^{N} Z_i \geq \frac{N}{2}) \\
&= \mathbb{P}(\frac{1}{N} \sum_{i=1}^{N} (Z_i - p) \geq \frac{1}{2} - p) \\
&\leq e^{-2N(\frac{1}{2} - p)^2} \\
&\leq e^{-\frac{N}{8}},
\end{aligned}$$

where the first inequality holds by Hoeffding's inequality and the second inequality holds due to $p \leq \frac{1}{4}$. Set $\delta = e^{-\frac{N}{8}}$, we have, with $N = O(\log(1/\delta))$, with probability at least $1 - \delta$, it holds $|\hat{\mu}_{MoM} - \mu| \leq \epsilon$. $\qquad \square$

## A.8. Proof of Theorem 4.9

Here, we explain how Theorem 4.9 is derived. We first show how the Median-of-Means (MoM) estimator and data splitting technique can conveniently convert Lemma 4.7 to a version holds with high probability.

For step $h$, Algorithm 1 can output a solution $\hat{w}_h$ such that $\mathbb{E}[\ell_h^\pi(\hat{w}_h) - \ell_h^\pi(w_h^*)] \leq \frac{\epsilon^2}{32\xi H^4}$. We can apply Lemma 4.8 on our algorithm which means that we can run the algorithm for $N = O(\log(1/\delta))$ times. Hence, we will get $N$ solutions $\{\hat{w}_{h,1}, \hat{w}_{h,2}, \ldots, \hat{w}_{h,N}\}$. Set $\hat{w}_{h,MoM}$ as the solution such that $\ell_h^\pi(\hat{w}_{h,MoM}) = \text{Median}(\ell_h^\pi(\hat{w}_{h,1}), \ell_h^\pi(\hat{w}_{h,2}), \ldots, \ell_h^\pi(\hat{w}_{h,N}))$. Based on Lemma 4.8, we have that with probability at least $1 - \delta$, it holds $\ell_h^\pi(\hat{w}_{h,MoM}) - \ell_h^\pi(w_h^*) \leq \frac{\epsilon^2}{32\xi H^4}$. With a little abuse of notation, we just denote $\hat{w}_{h,MoM}$ by $\hat{w}_h$ in the following content.

Now we are ready to estimate the total expected rewards of target policies, With the importance weighting ratio estimator $\frac{\hat{w}_h(s,a)}{\hat{\mu}_h(s,a)}$ from Algorithm 1, we can estimate the performance of policy $\pi^k$,

$$\hat{V}_1^{\pi^k} = \frac{1}{n} \sum_{i=1}^{n} \sum_{h=1}^{H} \frac{\hat{w}_h^{\pi^k}(s_h^i, a_h^i)}{\hat{\mu}_h(s_h^i, a_h^i)} r_h(s_h^i, a_h^i), \tag{17}$$

where $\{s_h^i, a_h^i\}_{i=1}^n$ is sampled from $\tilde{\mu}_h$.

**Lemma A.5.** *With samples $n = \tilde{O}\left(\frac{H^2}{\epsilon^2} \sum_{h=1}^{H} \max_{k \in [K]} \sum_{s,a} \frac{(\hat{d}_h^{\pi^k}(s,a))^2}{\hat{\mu}_h(s,a)}\right)$, we have with probability at least $1 - \delta$, $|\hat{V}_1^{\pi^k} - V_1^{\pi^k}| \leq \frac{\epsilon}{2}$, $k \in [K]$.*

*Proof.* First, we can decompose the error $|\hat{V}_1^{\pi^k} - V_1^{\pi^k}| = |\hat{V}_1^{\pi^k} - \mathbb{E}[\hat{V}_1^{\pi^k}] + \mathbb{E}[\hat{V}_1^{\pi^k}] - V_1^{\pi^k}| \leq |\hat{V}_1^{\pi^k} - \mathbb{E}[\hat{V}_1^{\pi^k}]| + |\mathbb{E}[\hat{V}_1^{\pi^k}] - V_1^{\pi^k}|$. Then, by Bernstein's inequality, with samples $n = \tilde{O}\left(\frac{H^2}{\epsilon^2} \sum_{h=1}^{H} \max_{k \in [K]} \sum_{s,a} \frac{(\hat{d}_h^{\pi^k}(s,a))^2}{\hat{\mu}_h(s,a)}\right)$, we have, $|\hat{V}_1^{\pi^k} - \mathbb{E}[\hat{V}_1^{\pi^k}]| \leq \frac{\epsilon}{4}$. Based Lemma 4.7, we have, $|\mathbb{E}[\hat{V}_1^{\pi^k}] - V_1^{\pi^k}| \leq \frac{\epsilon}{4}$. $\qquad \square$

Remember that in Section 4.1, we ignore those states and actions with low estimated visitation distribution for each target policy which induce at most $\frac{\epsilon}{2}$ error. Combined with Lemma A.5, our estimator $\hat{V}_1^{\pi^k}$ finally achieves that with probability at least $1 - \delta$, $|\hat{V}_1^{\pi^k} - V_1^{\pi^k}| \leq \epsilon, k \in [K]$.

And for sample complexity, in our algorithm, we need to sample data in three procedures. First, for the coarse estimation of the visitation distribution, we need $\tilde{O}(\frac{1}{\epsilon})$ samples. Second, to estimate the importance-weighting ratio, we need

samples $\tilde{O}\left(\frac{H^4}{\epsilon^2}\sum_{h=1}^{H}\max_{k\in[K]}\sum_{s,a}\frac{(d_h^{\pi^k}(s,a))^2}{\mu_h^*(s,a)}\right)$. Last, to build the final performance estimator (9), we need samples $\tilde{O}\left(\frac{H^2}{\epsilon^2}\sum_{h=1}^{H}\max_{k\in[K]}\sum_{s,a}\frac{(\hat{d}_h^{\pi^k}(s,a))^2}{\hat{\mu}_h(s,a)}\right)$. Therefore, the total trajectories needed,

$$n = \tilde{O}\left(\frac{H^4}{\epsilon^2}\sum_{h=1}^{H}\max_{k\in[K]}\sum_{s,a}\frac{(d_h^{\pi^k}(s,a))^2}{\mu_h^*(s,a)}\right).$$

Moreover, notice that,

$$\max_{k\in[K]}\sum_{s,a}\frac{(\hat{d}_h^{\pi^k}(s,a))^2}{\hat{\mu}_h(s,a)} \le \max_{k\in[K]}\sum_{s,a}\frac{(\hat{d}_h^{\pi^k}(s,a))^2}{\mu_h^*(s,a)} \le \frac{25}{16}\sum_{s,a}\frac{(d_h^\pi(s,a))^2}{\mu_h^*(s,a)}, \tag{18}$$

where $\mu_h^*$ is the optimal solution of the optimization problem (4), the first inequality holds due to $\hat{\mu}_h$ is the minimum of the approximate optimization problem (5) and the second inequality holds due to $\hat{d}_h^\pi(s,a) \le \frac{5}{4}d_h^\pi(s,a)$. Based on (18), we can substitute the coarse distribution estimator in the sample complexity bound by the exact one,

$$n = \tilde{O}\left(\frac{H^4}{\epsilon^2}\sum_{h=1}^{H}\max_{k\in[K]}\sum_{s,a}\frac{(d_h^{\pi^k}(s,a))^2}{\mu_h^*(s,a)}\right).$$

### A.9. Proof of Corollary 4.10

Let the sampling distribution $\mu_h'$ be $\frac{1}{SA}\sum_{s,a}d_h^{\pi_{s,a}}$, where $\pi_{s,a} = \arg\max_{k\in[K]}d_h^{\pi^k}(s,a)$. Since $\mu_h^*$ is the optimal solution and $\mu_h'$ is a feasible solution, we have $\forall h \in [H]$,

$$\max_{k\in[K]}\sum_{s,a}\frac{(d_h^{\pi^k}(s,a))^2}{\mu_h^*(s,a)} \le \max_{k\in[K]}\sum_{s,a}\frac{(d_h^{\pi^k}(s,a))^2}{\mu_h'(s,a)}$$
$$\le SA.$$

Notice that there is a logarithm term $\log(K)$ hidden in $\tilde{O}$ notation. Remember $K$ is the number of target policies. If they are deterministic, then $K$ is bounded by $A^{SH}$ which leads to $\log(K) \le SH\log(A)$.

Together, we have the sample complexity is bounded by $\tilde{O}\left(\frac{\text{poly}(H)S^2A}{\epsilon^2}\right)$.

# B. Lower order coarse estimation

---

**Algorithm 3** **M**ulti-policy **A**pproximation via **R**atio-based **C**oarse **H**andling (MARCH)

**Input:** Horizon $H$, accuracy $\epsilon$, policy $\pi$.

Coarsely estimate $d_1$ such that $dist^\beta(\hat{d}_1, d_1) \le \epsilon$, where $\beta = \frac{1}{H}$.

**for** $h = 1$ **to** $H - 1$ **do**

    1. Coarsely estimate $\mu_h$ such that $|\hat{\mu}_h(s,a) - \mu_h(s,a)| \le \max\{\epsilon', c \cdot \mu_h(s,a)\}$, where $\epsilon' = \frac{\epsilon}{2H^2 S^2 A^2}$ and $c = \frac{\beta}{2}$.

    2. Sample $\{s_h^i, a_h^i, s_{h+1}^i\}_{i=1}^n$ from $\mu_h$.

    3. Estimate $d_{h+1}(s,a)$ by $\hat{d}_{h+1}(s,a) = \frac{1}{n}\sum_{i=1}^n \mathbb{I}(s_{h+1}^i = s)\hat{w}_h(s_h^i, a_h^i)$.

**end for**

**Output:** $\{\hat{d}_h\}_{h=1}^H$.

---

In this section, we first provide our algorithm MARCH for coarse estimation of all the deterministic policies and then conduct an analysis on its sample complexity.

MARCH is based on the algorithm EULER proposed by Zanette and Brunskill (2019).

**Lemma B.1** (Theorem 3.3 in Jin et al. (2020))**.** *Based on* EULER, *with sample complexity $\tilde{O}(\frac{poly(H,S,A)}{\epsilon})$, we can construct a policy cover which generates a dataset with the distribution $\mu$ such that, with probability $1 - \delta$, if $d_h^{max}(s) \ge \frac{\epsilon}{SA}$, then,*

$$\mu_h(s,a) \ge \frac{d_h^{max}(s,a)}{2HSA}, \tag{19}$$

*where $d_h^{max}(s) = \max_\pi d_h^\pi(s), d_h^{max}(s,a) = \max_\pi d_h^\pi(s,a)$.*

With this dataset, we estimate the visitation distribution of deterministic policies by step-to-step importance weighting,

$$\hat{d}_{h+1}(s,a) = \frac{1}{n}\sum_{i=1}^n \mathbb{I}(s_{h+1}^i = s)\hat{w}_h(s_h^i, a_h^i),$$

where $\{s_h^i, a_h^i, s_{h+1}^i\}_{i=1}^n$ are sampled from $\mu$ and $\hat{w}_h(s,a) = \frac{\hat{d}_h(s,a)}{\hat{\mu}_h(s,a)}$.

We state that MARCH can coarsely estimate the visitation distributions of all the deterministic policies by just paying a lower-order sample complexity which is formalized in the following theorem.

**Theorem B.2.** *Implement Algorithm 3 with the number of trajectories $n = \tilde{O}(\frac{poly(H,S,A)}{\epsilon})$, with probability at least $1 - \delta$, it holds that for any deterministic policy $\pi$,*

$$|\hat{d}_h^\pi(s,a), d_h^\pi(s,a)| \le \max\{\epsilon, \frac{d_h^\pi(s,a)}{4}\}, \ \forall s \in \mathcal{S}, a \in \mathcal{A}, h \in [H],$$

*where $\hat{d}^\pi$ is the distribution estimator.*

*Proof.* Our analysis is based a notion of distance defined in the following.

**Definition B.3** ($\beta-$distance)**.** For $x, y \ge 0$, we define the $\beta-$distance as,

$$dist^\beta(x,y) = \min_{\alpha \in [\frac{1}{\beta}, \beta]} |\alpha x - y|.$$

Correspondingly, for $x, y \in \mathbb{R}^n$,

$$dist^\beta(x,y) = \sum_{i=1}^n dist^\beta(x_i, y_i).$$

Based on its definition, we show in the following lemma that $\beta-$distance has some properties.

**Lemma B.4.** *The $\beta-$distance possesses the following properties for $(x, y, z, \gamma \geq 0)$:*

$$1.\ dist^\beta(\gamma x, \gamma y) = \gamma dist^\beta(x, y); \tag{20}$$
$$2.\ dist^\beta(x_1 + x_2, y_1 + y_2) \leq dist^\beta(x_1, y_1) + dist^\beta(x_2, y_2); \tag{21}$$
$$3.\ dist^{\beta_1 \cdot \beta_2}(x, z) \leq dist^{\beta_1}(x, y) \cdot \beta_2 + dist^{\beta_2}(y, z). \tag{22}$$

*Proof.* See Appendix C.1. ◻

The following lemma shows that if we can control the $\beta-$distance between $\hat{x}, x$, then we can show $\hat{x}$ achieves the coarse estimation of $x$.

**Lemma B.5.** *Suppose $dist^{1+\beta}(x, y) \leq \epsilon$, then it holds that,*

$$|x - y| \leq \beta y + (1 + \frac{\beta}{1 + \beta})\epsilon \leq 2 \max\{(1 + \frac{\beta}{1 + \beta})\epsilon, \beta y\}.$$

*Proof.* See Appendix C.2. ◻

The logic of the analysis is to show the $\beta-$distance between $\hat{d}_h$ and $d_h$ can be bounded at each layer by induction. Then by Lemma B.5, we show $\{\hat{d}_h\}_{h=1}^H$ achieves coarse estimation.

Suppose at layer $h$, we have $\hat{d}_h$ such that $dist^{(1+\beta)^h}(\hat{d}_h, d_h) < \epsilon_h$ where $\beta = \frac{1}{H}$. For notation simplicity, we omit the superscript $\pi$. The analysis holds for any policy.

We use importance weighting to estimate $\hat{d}_{h+1}$,

$$\hat{d}_{h+1}(s, a) = \frac{1}{n} \sum_{i=1}^n \mathbb{I}(s_{h+1}^i = s)\pi(a|s)\hat{w}_h(s_h^i, a_h^i),$$

where $\hat{w}_h(s, a) = \frac{\hat{d}_h(s,a)}{\hat{\mu}_h(s,a)}$.

We also denote,

$$\overline{d}_{h+1}(s, a) = \mathbb{E}_{(s_h, a_h, s_{h+1}) \sim \mu_h}[\mathbb{I}(s_{h+1} = s)\hat{w}_h(s_h, a_h)].$$

By (22) in Lemma B.4, we have,

$$dist^{(1+\beta)^{h+2}}(\hat{d}_{h+1}, d_{h+1}) \leq \underbrace{dist^{(1+\beta)}(\hat{d}_{h+1}, \overline{d}_{h+1})(1 + \beta)^{h+1}}_{A} + \underbrace{dist^{(1+\beta)^{h+1}}(\overline{d}_{h+1}, d_{h+1})}_{B}. \tag{23}$$

Next, we show how we can bound these two terms $(A)$ and $(B)$. Note that for $(s, h)$ where $d_h^{max}(s) < \frac{\epsilon}{SA}$, the induced $\beta-$distance error is at most $\epsilon$. Therefore, we can just discuss state-action pairs which satisfy Lemma B.1.

**Bound of** $(A)$  We first show the following lemma tells us that the importance weighting is upper-bounded.

**Lemma B.6.** *Based on the definition of $\mu$, the importance weighting is upper bounded,*

$$w_h(s, a) = \frac{d_h(s, a)}{\mu_h(s, a)} \leq 2HSA \frac{d_h(s, a)}{d_h^{max}(s, a)} \leq 2HSA.$$

*Hence, we can clip $\hat{w}_h(s, a)$ at $2HSA$ such that $\hat{w}_h(s, a) \leq 2HSA$.*

Let's define the random variable $Z_{h+1}(s, a) = \mathbb{I}(s_{h+1} = s)\hat{w}_h(s_h, a_h)$, then $\hat{d}_{h+1}(s, a) = \frac{1}{n} \sum_{i=1}^n Z_{h+1}^i(s, a)$. Since $\hat{w}_h(s_h, a_h)$ is bounded by Lemma B.6, we have,

$$\mathbb{V}[Z_{h+1}(s, a)] \leq \mathbb{E}[Z_{h+1}(s, a)^2] \leq 2HSA\mathbb{E}[Z_{h+1}(s, a)].$$

By Berstein's inequality, we have with probability at least $1 - \delta$,

$$|\hat{d}_{h+1}(s,a) - \mathbb{E}[\hat{d}_{h+1}(s,a)]| \leq \sqrt{\frac{2\mathbb{V}[Z_{h+1}(s,a)]\log(1/\delta)}{n}} + \frac{2HSA\log(1/\delta)}{3n}$$

$$\leq \sqrt{\frac{4HSA\mathbb{E}[\hat{d}_{h+1}(s,a)]\log(1/\delta)}{n}} + \frac{2HSA\log(1/\delta)}{3n},$$

to achieve the estimation accuracy $|\hat{d}_{h+1}(s,a) - \mathbb{E}[\hat{d}_{h+1}(s,a)]| \leq \max\{\epsilon, c \cdot \mathbb{E}[\hat{d}_{h+1}(s,a)]\}$, we need samples $n = \tilde{O}\left(\frac{HSA}{c \cdot \epsilon}\right)$.

Based on the above analysis, we can achieve,

$$|\hat{d}_{h+1}(s,a), \overline{d}_{h+1}(s,a)| \leq \max\{\epsilon', \frac{\beta}{2}\overline{d}_{h+1}(s,a)\}$$

at the cost of samples $\tilde{O}\left(\frac{HSA}{\beta\epsilon'}\right)$.

We now show $dist^{1+\beta}(\hat{d}_{h+1}, \overline{d}_{h+1}) \leq SA\epsilon'$. We discuss it in two cases,

$$1. \ |\hat{d}_{h+1}(s,a), \overline{d}_{h+1}(s,a)| \leq \epsilon' \tag{24}$$

$$2. \ |\hat{d}_{h+1}(s,a), \overline{d}_{h+1}(s,a)| \leq \frac{\beta}{2}\overline{d}_{h+1}(s,a). \tag{25}$$

For those $(s,a)$ which satisfies (25), since $[1 - \frac{\beta}{2}, 1 + \frac{\beta}{2}] \in [\frac{1}{1+\beta}, 1 + \beta]$, by the definition of $\beta-$distance, we have,

$$dist^{1+\beta}(\hat{d}_{h+1}(s,a), \overline{d}_{h+1}(s,a)) = 0. \tag{26}$$

For other $(s,a)$ which satisfies (24), we have,

$$dist^{1+\beta}(\hat{d}_{h+1}(s,a), \overline{d}_{h+1}(s,a)) \leq |\hat{d}_{h+1}(s,a), \overline{d}_{h+1}(s,a)| \leq \epsilon'.$$

Since there are at most $SA$ state-action pairs, the error in the second case is at most $SA\epsilon'$. Combine these two cases, we have,

$$dist^{1+\beta}(\hat{d}_{h+1}, \overline{d}_{h+1}) \leq SA\epsilon'.$$

By setting $\epsilon = \frac{\epsilon'}{SA}$, we have,

$$(A) = dist^{1+\beta}(\hat{d}_{h+1}, \overline{d}_{h+1})(1 + \beta)^{h+1} \leq (1 + \beta)^{h+1}\epsilon, \tag{27}$$

and the sample complexity is $\tilde{O}\left(\frac{(HSA)^2}{\epsilon}\right)$.

**Bound of** $(B)$  Next we show how to bound term $(B)$. Denote $\mu_h(s,a)\frac{\hat{d}_h(s,a)}{\hat{\mu}_h(s,a)}$ by $\tilde{d}_h(s,a)$, we have,

$$(B) = dist^{(1+\beta)^{h+1}}(\overline{d}_{h+1}, d_{h+1})$$

$$= \sum_{s,a} dist^{(1+\beta)^{h+1}}(\overline{d}_{h+1}(s,a), d_{h+1}(s,a))$$

$$= \sum_{s,a} dist^{(1+\beta)^{h+1}}(\sum_{s',a'} P_h^\pi(s,a|s',a')\tilde{d}_h(s',a'), \sum_{s',a'} P_h^\pi(s,a|s',a')d_h(s',a'))$$

$$\leq \sum_{s,a}\sum_{s',a'} dist^{(1+\beta)^{h+1}}(P_h^\pi(s,a|s',a')\tilde{d}_h(s',a'), P_h^\pi(s,a|s',a')d_h(s',a'))$$

$$= \sum_{s,a}\sum_{s',a'} P_h^\pi(s,a|s',a')dist^{(1+\beta)^{h+1}}(\tilde{d}_h(s',a'), d_h(s',a'))$$

$$= dist^{(1+\beta)^{h+1}}(\tilde{d}_h, d_h),$$

where the first two equality holds by definition, the inequality holds by (21) in Lemma B.4, the third equality holds by (20) in Lemma B.4 and the last one holds by $\sum_{s,a} P_h^\pi(s,a|s',a') = 1$.

Now we analyse $dist^{(1+\beta)^{h+1}}(\tilde{d}_h, d_h)$.

$$dist^{(1+\beta)^{h+1}}(\tilde{d}_h, d_h) = \sum_{s,a} \mu_h(s,a) dist^{(1+\beta)^{h+1}}(\frac{\hat{d}_h(s,a)}{\hat{\mu}_h(s,a)}, \frac{d_h(s,a)}{\mu_h(s,a)}).$$

By coarse estimation, we have $|\hat{\mu}_h(s,a) - \mu_h(s,a)| \le \max\{\epsilon', c \cdot \mu_h(s,a)\}$. Similarly, we discuss it in two cases,

$$1. |\hat{\mu}_h(s,a), \mu_h(s,a)| \le \epsilon', \tag{28}$$
$$2. |\hat{\mu}_h(s,a), \mu_h(s,a)| \le c \cdot \mu_h(s,a). \tag{29}$$

For those $(s,a)$ which satisfies (28), by Lemma B.6, we have,

$$dist^{(1+\beta)^{h+1}}(\frac{\hat{d}_h(s,a)}{\hat{\mu}_h(s,a)}, \frac{d_h(s,a)}{\mu_h(s,a)}) \le |\frac{\hat{d}_h(s,a)}{\hat{\mu}_h(s,a)} - \frac{d_h(s,a)}{\mu_h(s,a)}| \le 2HSA.$$

Hence, we have,

$$dist^{(1+\beta)^{h+1}}(\tilde{d}_h(s,a), d_h(s,a)) = \mu_h(s,a) dist^{(1+\beta)^{h+1}}(\frac{\hat{d}_h(s,a)}{\hat{\mu}_h(s,a)}, \frac{d_h(s,a)}{\mu_h(s,a)})$$
$$\le 2HSA\mu_h(s,a) \le \frac{2HSA\epsilon'}{c},$$

where the last inequality holds by $c \cdot \mu_h(s,a) \le \epsilon'$.

Next, For those $(s,a)$ which satisfies (29), we have,

$$(1-c)\frac{1}{\hat{\mu}_h(s,a)} \le \frac{1}{\mu_h(s,a)} \le (1+c)\frac{1}{\hat{\mu}_h(s,a)}.$$

Set $c = \frac{\beta}{2}$, since $[1 - \frac{\beta}{2}, 1 + \frac{\beta}{2}] \in [\frac{1}{1+\beta}, 1+\beta]$, by definition of $\beta-$distance, we have,

$$dist^{(1+\beta)}(\frac{1}{\hat{\mu}_h(s,a)}, \frac{1}{\mu_h(s,a)}) = 0. \tag{30}$$

And we assume by induction that $dist^{(1+\beta)^h}(\hat{d}_h(s,a), d_h(s,a)) \le \epsilon_h$, together with (30) we have,

$$dist^{(1+\beta)^{h+1}}(\frac{\hat{d}_h(s,a)}{\hat{\mu}_h(s,a)}, \frac{d_h(s,a)}{\mu_h(s,a)}) \le \epsilon_h. \tag{31}$$

Combine the results of two cases together, we have,

$$(B) = dist^{(1+\beta)^{h+1}}(\tilde{d}_h, d_h) \le \epsilon_h + 4H^2 S^2 A^2 \epsilon'$$

Set $\epsilon' = \frac{\epsilon}{4H^2 S^2 A^2}$, we have,

$$(B) \le \epsilon_h + \epsilon \tag{32}$$

at the cost of samples $\tilde{O}(\frac{H^3 S^2 A^2}{\epsilon})$.

Now we are ready to show the bound of $\beta-$distance at layer $h+1$. Plug (27)(32) into (23), we have,

$$dist^{(1+\beta)^{h+2}}(\hat{d}_{h+1}, d_{h+1}) \le dist^{(1+\beta)}(\hat{d}_{h+1}, \overline{d}_{h+1})(1+\beta)^{h+1} + dist^{(1+\beta)^{h+1}}(\overline{d}_{h+1}, d_{h+1})$$
$$\le (1+\beta)^{h+1}\epsilon + \epsilon + \epsilon_h.$$

Start from $dist^{(1+\beta)}(\hat{d}_1, d_1) \leq \epsilon$, we have,

$$dist^{(1+\beta)^{2h-1}}(\hat{d}_h, d_h) \leq h\epsilon + \epsilon \sum_{l=1}^{h-1} (1+\beta)^{2h}. \tag{33}$$

Remember that $\beta = \frac{1}{H}$ and due to $(1 + \frac{1}{H})^h \leq e \ (h \leq H)$, we have,

$$dist^{e^2}(\hat{d}_h, d_h) \leq H(1+e^2)\epsilon. \tag{34}$$

Recall Lemma B.5, and based on (34), we have,

$$|\hat{d}_h(s,a) - d_h(s,a)| \leq 2\max\{H(1+e^2)\epsilon, (e^2-1)d_h(s,a)\}.$$

By just paying multiplicative constant, we can adjust the constant above to meet our needs, i.e. in Theorem B.2. □

# C. Proof of lemmas in Section B

## C.1. Proof of Lemma B.4

*Proof.* 1. The first property is trivial.

$$dist^\beta(\gamma x, \gamma y) = \min_{\alpha \in [\frac{1}{\beta}, \beta]} |\alpha \gamma x - \gamma y|$$
$$= \min_{\alpha \in [\frac{1}{\beta}, \beta]} \gamma |\alpha x - y|$$
$$= \gamma dist^\beta(x, y).$$

2. Let $\alpha_i$ be such that,

$$dist^{1+\beta}(x_i, y_i) = |\alpha_i x_i - y_i|, \ i = 1, 2.$$

Notice that $\alpha_3 = \alpha_1 \cdot \frac{x_1}{x_1+x_2} + \alpha_2 \cdot \frac{x_2}{x_1+x_2}$ satisfies $\alpha_3 \in [\alpha_1, \alpha_2] \in [\frac{1}{\beta}, \beta]$ and $\alpha_3(x_1 + x_2) = \alpha_1 x_1 + \alpha_2 x_2$, therefore,

$$dist^\beta(x_1 + x_2, y_1 + y_2) = \min_{\alpha \in [\frac{1}{\beta}, \beta]} |\alpha(x_1 + x_2) - y_1 - y_2|$$
$$\leq |\alpha_3(x_1 + x_2) - y_1 - y_2|$$
$$= |\alpha_1 x_1 + \alpha_2 x_2 - y_1 - y_2|$$
$$\leq |\alpha_1 x_1 - y_1| + |\alpha_2 x_2 - y_2|$$
$$= dist^\beta(x_1, y_1) + dist^\beta(x_2, y_2).$$

The first inequality holds due to the definition of $\beta-$distance. The second inequality is the triangle inequality.

3. We prove the third property through a case-by-case discussion.

(1). $\frac{x}{\beta_1 \beta_2} \leq z \leq \beta_1 \beta_2 x$. In this case, the result is trivial, since $dist^{\beta_1 \beta_2}(x, z) = 0$ and $\beta-$distance is always non-negative.

(2). $\beta_1 \beta_2 x < z$. If $y \leq x$, then,

$$dist^{\beta_1 \beta_2}(x, z) \leq dist^{\beta_2}(x, z) \leq dist^{\beta_2}(y, z).$$

We are done.

If $x < y \leq \beta_1 x$, then $dist_1^\beta(x, y) = 0$, and $z > \beta_1 \beta_2 x \geq \beta_2 y$, hence,

$$dist^{\beta_2}(y, z) = z - \beta_2 y \geq z - \beta_1 \beta_2 x = dist^{\beta_1 \beta_2}(x, z).$$

We are done.

If $y > \beta_1 x, z \in [\frac{y}{\beta_2}, \beta_2 y]$, then,

$$dist^{\beta_1}(x, y)\beta_2 + dist^{\beta_2}(y, z) = \beta_2(y - \beta_1 x)$$
$$\geq z - \beta_1 \beta_2 x$$
$$= dist^{\beta_1 \beta_2}(x, z).$$

We are done.

If $y > \beta_1 x, z \notin [\frac{y}{\beta_2}, \beta_2 y]$, then,

$$dist^{\beta_1}(x, y)\beta_2 + dist^{\beta_2}(y, z) \geq \beta_2(y - \beta_1 x)$$
$$\geq z - \beta_1 \beta_2 x$$
$$= dist^{\beta_1 \beta_2}(x, z).$$

We are done.

(3). $z < \frac{x}{\beta_1\beta_2}$. A symmetric analysis can be done by replacing $\beta_1, \beta_2$ by $\frac{1}{\beta_1}, \frac{1}{\beta_2}$ which gives the result,

$$dist^{\beta_1\beta_2}(x,z) \le dist^{\beta_1}(x,y)\frac{1}{\beta_2} + dist^{\beta_2}(y,z)$$

Since $\beta_2 \ge 1$ and $dist^{\beta_1}(x,y) \ge 0$, we have $dist^{\beta_1}(x,y)\frac{1}{\beta_2} \le dist^{\beta_1}(x,y)\beta_2$, hence,

$$dist^{\beta_1\beta_2}(x,z) \le dist^{\beta_1}(x,y)\beta_2 + dist^{\beta_2}(y,z),$$

which concludes the proof. $\qquad\square$

### C.2. Proof of Lemma B.5

*Proof.* We prove the lemma through a case-by-case study.

(1). $x \le y$. If $dist^{1+\beta}(x,y) = 0$, then $x(1+\beta) \ge y \ge x$, therefore,

$$|x - y| = y - x \le \beta x \le \beta y.$$

If $dist^{1+\beta}(x,y) > 0$, then $dist^{1+\beta}(x,y) = y - (1+\beta)x$, therefore,

$$|x - y| = y - x = dist^{1+\beta}(x,y) + \beta x \le \epsilon + \beta x \le \epsilon + \beta y.$$

(2). $y < x$. If $dist^{1+\beta}(x,y) = 0$, then $\frac{x}{1+\beta} \le y < x$, therefore,

$$|x - y| = x - y \le x - \frac{x}{1+\beta} \le y(1+\beta)(1 - \frac{1}{1+\beta}) = \beta y.$$

If $dist^{1+\beta}(x,y) > 0$, then $y < \frac{x}{1+\beta} \le x$ and $dist^{1+\beta}(x,y) = \frac{x}{1+\beta} - y$. Moreover, since $dist^{1+\beta}(x,y) \le \epsilon$, we have $\frac{x}{1+\beta} \le \epsilon + y$. Therefore,

$$\begin{aligned}
|x - y| &= x - y \\
&= dist^{1+\beta}(x,y) + (1 - \frac{1}{1+\beta})x \\
&= dist^{1+\beta}(x,y) + \beta\frac{x}{1+\beta} \\
&\le \epsilon + \frac{\beta}{1+\beta}\epsilon + \beta y \\
&= (1 + \frac{\beta}{1+\beta})\epsilon + \beta y.
\end{aligned}$$

Combine the results above together, we have,

$$|x - y| \le \beta y + (1 + \frac{\beta}{1+\beta})\epsilon \le 2\max\{(1 + \frac{\beta}{1+\beta})\epsilon, \beta y\}.$$

$\qquad\square$

# D. Discussion on policy identification

In this section, we discuss on the application of CAESAR to policy identification problem, its instance-dependent sample complexity and some intuitions related to the existing gap-dependent results.

We first provide a simple algorithm that utilizes CAESAR to identify an $\epsilon-$optimal policy. The core idea behind the algorithm is we can use CAESAR to evaluate all candidate policies up to an accuracy, then we can eliminate those policies with low estimated performance. By decreasing the evaluation error gradually, we can finally identify a near-optimal policy with high probability.

For notation simplicity, fixing the high-probability factor, we denote the sample complexity of CAESAR by $\frac{\Theta(\Pi)}{\gamma^2}$, where $\Pi$ is the set of policies to be evaluated and $\gamma$ is the estimation error.

---

**Algorithm 4** Policy Identification based on CAESAR

    **Input:** Alg CAESAR , optimal factor $\epsilon$, candidate policy set $\Pi$.
    **for** $i = 1$ **to** $\lceil \log_2(4/\epsilon) \rceil$ **do**
        1. Run CAESAR to evaluate the performance of policies in $\Pi$ up to accuracy $\gamma = \frac{1}{2^i}$.
        2. Eliminate $\pi^i$ if $\exists \pi^j \in \Pi, \hat{V}_1^{\pi^j} - \hat{V}_1^{\pi^i} > 2\gamma$, update $\Pi$.
    **end for**
    **Output:** Randomly pick $\pi^o$ from $\Pi$.

---

**Theorem D.1.** *Implement Algorithm 4, we have that, with probability at least $1 - \delta$, $\pi^o$ is $\epsilon-$optimal, i.e.,*

$$V_1^* - V_1^{\pi^o} \leq \epsilon.$$

*And the instance-dependent sample complexity is $\tilde{O}(\max_{\gamma \geq \epsilon} \frac{\Theta(\Pi_\gamma)}{\gamma^2})$, where $\Pi_\gamma = \{\pi : V_1^* - V_1^\pi \leq 8\gamma\}$.*

*Proof.* On the one hand, based on the elimination rule in the algorithm, by running CAESAR with the evaluation error $\gamma$, the optimal policy $\pi^*$ will not be eliminated with probability at least $1-\delta$. Since $\max_{\pi \in \Pi} \hat{V}_1^\pi - \hat{V}_1^{\pi^*} \leq V_1^* + \gamma - (V_1^{\pi^*} - \gamma) \leq 2\gamma$.

On the other hand, if $V_1^* - V_1^{\pi^i} > 4\gamma$, then $\pi^i$ will be eliminated with probability at least $1 - \delta$. Since $\max_{\pi \in \Pi} \hat{V}_1^\pi - \hat{V}_1^{\pi^i} > V_1^* - \gamma - (V_1^{\pi^i} + \gamma) > 2\gamma$.

Therefore, by running Algorithm 4, the final policy set is not empty and for any policy $\pi$ in this set, it holds, $V_1^* - V_1^\pi \leq \epsilon$ with probability at least $1 - \delta$.

Next, we analyse the sample complexity of Algorithm 4. Based on above analysis, within every iteration of the algorithm, we have a policy set containing $8\gamma-$optimal policies, and we use CAESAR to evaluate the performance of these policies up to $\gamma$ accuracy. By Theorem 4.9, the sample complexity is $\frac{\Theta(\Pi_\gamma)}{\gamma^2}$. Therefore, the overall sample complexity is,

$$\sum_\gamma \frac{\Theta(\Pi_\gamma)}{\gamma^2} \leq \tilde{O}(\max_{\gamma \geq \epsilon} \frac{\Theta(\Pi_\gamma)}{\gamma^2}).$$

$\square$

This result is quite interesting since it provides another perspective beyond the existing gap-dependent results for policy identification. And these two results have some intuitive relations that may be of interest.

Roughly speaking, to identify an $\epsilon-$optimal policy for an MDP, the gap-dependent regret is described as,

$$O(\sum_{h,s,a} \frac{H \log K}{gap_h(s,a)}),$$

where $gap_h(s, a) = V_h^*(s) - Q_h^*(s, a)$.

The value gap $gap_h(s, a)$ quantifies how sub-optimal the action $a$ is at state $s$. If the gap is small, it is difficult to distinguish and eliminate the sub-optimal action. At the same time, smaller gaps mean that there are more policies with similar

performance to the optimal policy, i.e. the policy set $\Pi_\gamma$ is larger. Both our result and gap-dependent result can capture this intuition. We conjecture there exists a quantitative relationship between these two perspectives.

An interesting proposition of Theorem D.1 is to apply the same algorithm to the multi-reward setting. A similar instance-dependent sample complexity can be achieved $\tilde{O}(\max_{\gamma \geq \epsilon} \frac{\Theta(\Pi_\gamma^{\mathcal{R}})}{\gamma^2})$ with the difference that $\Pi_\gamma^{\mathcal{R}}$ contains policies which is $8\gamma-$optimal for at least one reward function. This sample complexity captures the intrinsic difficulty of the problem by how similar the near-optimal policies under different rewards are which is consistent with the intuition.

