# OpenReview forum: "Multiple-policy Evaluation via Density Estimation"
_ICML.cc/2025/Conference — ICML 2025 poster_

### Official Review · Reviewer_Qkrf · 2025-02-28

**Overall Recommendation:** 3

**Summary:**

The paper introduces CAESAR, an algorithm for efficiently evaluating multiple policies in finite-horizon MDPs to a desired accuracy and confidence level. CAESAR first obtains coarse estimates of the visitation distributions with low sample complexity, and then refines these estimates by computing importance weights using a step-wise quadratic loss inspired by DualDICE. Additional contributions include the IDES subroutine for improved importance ratio estimation and the MARCH algorithm, which, along with a new β-distance metric, enables efficient estimation even with exponentially many deterministic policies.

**Claims And Evidence:**

The theoretical results is  backboned by comprehensive proof.

**Essential References Not Discussed:**

I think this paper is widely related to offline RL (instance-dependence bound), offline policy evaluation and I think the paper should add more dicussions related to these literature.

**Experimental Designs Or Analyses:**

No experiments.

**Methods And Evaluation Criteria:**

The result is evaluated in terms of sample complexity.

**Other Comments Or Suggestions:**

- Add more clarifications on the setting and the definition
- Since a new concept/measure is proposed, it would be better to use some experiment to show whether it's reasonable or not.

**Other Strengths And Weaknesses:**

- The paper is well-organized and the main idea of the whole algorithm is well devided into several sections with detailed discussions
- The idea is quite new and I think the problem fomulation itself in practice since it may be related to how to construct a shared dataset for policy evaluation
- However, I'm quite confusion with the setting since it seems to be hibrid evaluation process since the coarse  estimation is estimated online and then the policy is evaluted, in an offline pattern, with a dataset collected by the chosen sampling policy based on the estimation attained online. Such a hybrid process is quite counteriintuitive to me and need more clarifications.
- Also, the definition of coarse estimation seems new and I'm not sure whether it's reasonable.

**Questions For Authors:**

- Please clarify the weakness part.
- Could author connect the proposed method more with offline RL literature and show what's the similarity and difference since the high-level idea seems really similar to me
- The comparison with existing results is unclear to me which is better. Would it be possible to provide a more clear result in a specific setting?
- Why MoM is introduced? Why the bouned expectation in Lemma 4.7 does not directly indicate the high probability bound?
- Why lower bound of MARCH need the assumption that policies are deterministic? From my intepretation, it's a result for tabular MDP

**Relation To Broader Scientific Literature:**

see below

**Theoretical Claims:**

I checked the prooffor Secton 4.1 and 4.2 briefly and it seems make sense.

---

> ### Author Rebuttal · Authors · 2025-03-31
>
> Thank you for your efforts on reviewing our paper. We are confident that we can solve your concerns and questions. And we kindly request you to consider increasing your score if you think we address your concerns.
>
> > The hybrid setting of online and offline is confusing.
>
> Sorry to cause the confusion. The setting of our problem is online. Our problem formulation is: given a set of target policies and an unknown MDP, evaluate the performance of all these policies. No additional dataset is given, interactions with the environment are necessary, so it is definitely online.
>
> We do mention in the paper that our method is implemented more like in an offline manner. Since for offline RL, an offline dataset is given and no more interaction is allowed, i.e. we can only sample data from this fixed given distribution. This is exactly the second phase of our algorithm, where we have a fixed sampling distribution/policy derived from the first phase, and use data from this fixed distribution to perform the evaluation.
>
> > The connection between our method and offline RL.
>
> In offline RL, an offline dataset is given and usually is assumed to have good coverage over the state space. In our scenario, if we already have an offline dataset that has good coverage over the space of these target policies, then we can directly use methods from offline RL to do off-policy evaluation of these target policies. However, it is not the case. We don't assume such a good offline dataset which is always unpractical because we have multiple policies, and even if it exists, it may be super sub-optimal with respect to the policies at hand, e.g. a uniform offline dataset.
>
> What our method does is, we first interact with the environment to get some information of the target policies and then we can calculate  an approximately optimal sampling policy in terms of the coverage over the target policies. We successfully show that only low-order  samples are needed to construct such a sampling dataset/policy. In the second phase, we can just sample data using this fixed sampling policy to do policy evaluation just like offline RL.
>
> > Doubts on the reasonability of coarse estimation.
>
> That's one of our contributions. In our work, we theoretically show the effectiveness of coarse estimation in multiple-policy evaluation problem. The coarse estimation has been proved reasonable.
>
> We are not sure what exact concerns you have about coarse estimation. Are you concerned about how coarse estimation leads to final accurate performance evaluation? It is true that one can never achieve estimation up to $\epsilon-$accuracy by just doing coarse estimation with $\tilde{O(1/\epsilon)}$ sample costs, the technique is that coarse estimation is only used to approximate an intermediate quantity in the whole procedure, e.g. the sampling distribution $\hat \mu^*$ in our scenario. In our setting, to evaluate the final performance accurately, the second phase is unavoidable.
>
> Coarse estimation is powerful since first, it only needs low-order samples which is negligible compared to the main order of sample costs, second, the multiplicative constant error it induces is acceptable. In other words, it enables us to approximate some useful intermediate quantities at an acceptable sample cost and error which serves an important role to achieve the ultimate goal.
>
>
> > The comparison with existing results is unclear which is better.
>
> We already discussed the comparison of our sample complexity with the existing work in Section 5.1. It is unclear whether our sample complexity is universally better, however, our result is meaningful and significant by offering new insights for the problem and sharing many advantages. First, we solve the unfavorable dependency of $1/d^{max}(s)$ in existing results. Second, our work tackles the problem from a different perspective which is complementary to the existing method. Third, our sample complexity is cleaner and more  interpretable than the existing result. Finally, we can parallelize our method while the existing method can't since it relies on trajectory stitch which is a practical advantage.
>
> > Why lower bound of MARCH need the assumption that policies are deterministic?
>
> It is not an assumption. MARCH is our proposed algorithm to coarsely estimate all deterministic policies. Since we consider tabular MDPs, any policy can be expressed as a combination of deterministic policies. So it is enough to estimate all deterministic policies.
>
> > Why MoM is introduced? Why the bounded expectation in Lemma 4.7 does not directly indicate the high probability bound?
>
> A result holds in expectation usually does not imply that the result holds with high probability. In our scenario, we get our importance density estimators by Algorithm 1 which uses stochastic gradient descent.
>
> > Other comments
>
> Thanks for the comment. We will consider updating our paper to make the setting clearer, include more related works in offline RL and implement some experiments.

---

### Official Review · Reviewer_S332 · 2025-03-07

**Overall Recommendation:** 4

**Summary:**

This paper addresses the problem of evaluating the performance of multiple target policies in reinforcement learning (RL) using a sample-efficient algorithm called CAESAR. Existing methods for single-policy evaluation can be inefficient when applied separately to each policy, especially when policies are similar. CAESAR tackles this by first estimating coarse visitation distributions for all policies, then designing an approximately optimal sampling distribution that efficiently covers all target policies simultaneously. The sampled data is used to estimate policy values using importance weighting, with importance ratios estimated through a new technique called IDES, inspired by the DualDICE method. Compared to prior work, CAESAR offers a flexible offline evaluation framework, works for finite-horizon MDPs, and is applicable to both policy evaluation and near-optimal policy identification tasks.

**Claims And Evidence:**

The claims made in the paper are well supported with the theoretical results accompanied.

**Essential References Not Discussed:**

I am not aware of any essential references that are not discussed in this paper.

**Experimental Designs Or Analyses:**

The paper is not performing any experiments.

**Methods And Evaluation Criteria:**

The proposed methods has strong theoretical backgrounds. It is not empirically evaluated, but it is evaluated with its sample complexity and compared with other algorithms with its order.

**Other Comments Or Suggestions:**

.

**Other Strengths And Weaknesses:**

- very detailed, easy to follow proofs
- the paper is overall easy to follow
- deals with an issue that have a number of practical applications
- purely theoretic, no empirical analyses

**Questions For Authors:**

I am not sure whether framing the task as "offline multiple-policy evaluation" is appropriate, because at a first glance it looks like a trivial problem, evaluating multiple policies with a given offline dataset. IMO, it would be easier to understand if the task was framed as computing a policy with least OPE error on multiple policies.

**Relation To Broader Scientific Literature:**

- This paper contributes to the less explored area of what policy should be used to effectively evaluate multiple policies. This paper seems to be one of a very few papers that provides theoretical results on this field. This may have some impact to practical algorithms who have similar goals.
- This paper proposes IDES, a non-trivial extension of DualDICE to finite-horizon MDPs.

**Theoretical Claims:**

I was not able to fully check the correctness of theorems, but the proofs seem to be detailed enough to easily point out the issues, and I haven't found any issue.

---

> ### Author Rebuttal · Authors · 2025-04-01
>
> Thank you for your efforts on reviewing our paper. We really appreciate your positive feedbacks on our work. We agree that framing the task as 'offline' multiple-policy evaluation is kind of wired and may cause some unnecessary confusions. Thanks for pointing it out and we will consider reorganizing the language.

---

### Official Review · Reviewer_YEZA · 2025-03-14

**Overall Recommendation:** 3

**Summary:**

This paper tackles the problem of multiple-policy evaluation, by proposing an offline off-policy approach named “CAESAR”, in contrast to the online on-policy approach by Dann et al. 2023.

The proposed approach first performs a coarse estimate of the visitation distributions of the target policies. These estimates are then used to approximate the optimal sampling distribution and to compute the importance density ratio, drawing inspirations from DualDICE [Nachum et. al. 2019]. For estimating the visitation distributions of deterministic target polices, they also proposed an algorithm MARCH, leveraging a novel notion called $\beta$-distance.

Furthermore, the paper provides a detailed analysis of the sample complexity of CAESAR, presenting PAC bounds and show how they scale with key parameters such as horizon H, number of policies K, number of states S and actions A.

**Claims And Evidence:**

Yes

**Essential References Not Discussed:**

No

**Experimental Designs Or Analyses:**

There are no experiments in this paper.

**Methods And Evaluation Criteria:**

Yes

**Other Comments Or Suggestions:**

No.

**Other Strengths And Weaknesses:**

1. The paper introduced the notion $\beta$-distance for analyzing the sample complexity of MARCH. However, the intuition behind $\beta$-distance does not seem entirely clear. Additional discussions about that would enhance the clarity.
2. Due to the $H^4$ dependence of sample complexity, it would strengthen the paper to include experiments that empirically verify that scaling.

**Questions For Authors:**

The sample complexity in Theorem 4.9 scales with $H^4$, compared to the $H^2$ scaling in Dann et.al (2023) has $H^2$. This higher order dependence on $H$ may limit the practical applicability of the proposed method.  You mentioned that this gap was induced by the error propagation and conjectured that a different loss function might address it. Can you elaborate on that point and discuss if there are other approaches that might reduce the dependence on $H$?

**Relation To Broader Scientific Literature:**

It extends prior work in multiple-policy evaluation from online, on-policy setting to an offline off-policy setting. And it adapts and extends the distribution ratio estimation technique from DualDICE (Nachum et al., 2019) to the finite horizon context, with a step-wise objective function.

**Theoretical Claims:**

I checked the proof of the main results and did not identify obvious issues.

---

> ### Author Rebuttal · Authors · 2025-03-31
>
> Thank you for your efforts on reviewing our paper. We appreciate your comments. And we kindly request you to consider increasing your score if we solve your concerns and questions.
>
> > The intuition behind $\beta$-distance does not seem entirely clear.
>
> The $\beta$-distance is defined as follows $dist^{\beta}(x,y)=\min_{\alpha\in[1/\beta, \beta]} |\alpha x - y|$. The $\beta$-distance is zero as long as $\frac{x}{\beta} \le y \le \beta x$ which means $x$ approximates $y$ up to multiplicative constant $\beta$ error. The intuition behind $\beta$-distance is that we want a metric that can measure the coarse estimation. $\beta$-distance is such a metric. We showed that in  Lemma B.5, if $\beta$-distance is smaller than $\epsilon$, then we can show $x$ is a coarse estimator for $y$. With this metric, we can focus on analyzing the bound of $\beta$-distance, $dist^{\beta}(\hat d, d)$, to show $\hat d$ is a coarse estimator for visitation distribution $d$. Analyzing $\beta$-distance is much more convenient than analyzing the formulation in Definition 4.1.
>
> Thanks for the comment. The detailed elaboration of $\beta$-distance and MARCH is in Appendix B. We will consider updating the paper to clarify $\beta$-distance and MARCH more in the main page.
>
> > It would be better to include experiments that can show the $H^4$ dependency of sample complexity.
>
> We do agree with the reviewer that having such an experiment will strengthen the paper, but practically, to show the dependency of $H^4$ in experiments would be extremely hard since it is just an upper bound with high probability. And we also want to reiterate that we consider the scope of our submission to be the advancement of our theoretical understanding of this problem. As such we hope to obtain a fair evaluation of our submission based on its merits to advance theoretical research in RL.
>
> What's more, same as most theoretical works in RL, we were concerned with achieving the sharpest S and K (for this specific problem) dependence, which given the large size of S in applications is typically considered more consequential than dependence on H. Our minimax upper bounds achieve the sharpest dependence on S and A.
>
>
> > Elaborate that a different loss function might address the sub-optimal dependency on $H$. And is there any other approach to address  it?
>
> The loss function we defined to estimate the importance densities is a step-wise loss function (see L312-315 (right)). And we are adopting a step-by-step optimization procedure, i.e., we first get the estimator at step h-1 by minimizing the loss function $l_{h-1}(w)$, and then, utilizing it to get estimator at step $h$ by minimizing $l_h(w)$. Since the loss function at step $h$ depends on the estimator from last step, there is an additive error propagation which induces the $H^4$ dependency.
>
> What we are thinking is to propose a comprehensive loss function $L(w)$ that incorporates all steps utilizing some coarse knowledge of transition functions. By minimizing this loss function, we can get the importance density estimator for all layers at once which avoids the step-to-step error propagation thus gets rid of $H^4$ dependency.
>
> It is a good question that if there are other approaches to solve multiple policy evaluation problems well with optimal dependency on $H$. Based on our pipeline, we think it is hard to do more beyond the method we described above but we are optimistic that there exists totally different methods that can achieve it. Multiple policy evaluation is a challenge and under-studied problem which is worth exploring.

---

### Official Review · Reviewer_HUdG · 2025-03-14

**Overall Recommendation:** 2

**Summary:**

Propose algorithm for off-policy estimation of a set of $K$ policies. To me, the main idea is to try to improve the linear dependence in $K$ in the naive $K / \varepsilon^2$ sample complexity (where one simply estimates the visitations of all $K$ policies via Monte Carlo) by collecting samples from a covering distribution, then doing importance-weighted estimation.

The method is comprised of the following steps (Algorithm 2)
- Estimate the visitation distributions of each policy up to constant accuracy -- the sample complexity here has a fast statistical rate $\varepsilon^{-1}$, but I believe it scales as $\text{poly}(K,S,A)$, see below
- Compute the "optimal sampling distribution" $\hat\mu_h^*$, which is essentially the distribution that best covers the estimated distributions of the $K$ policies (in the sense of minimizing variance or expected coverage, aka $\ell_1$ or $\ell_2$-coverability ), and collect samples
- Since the data is now off-policy, importance weights must be computed in order to estimate the policies' values (Algorithm 1)

**Claims And Evidence:**

Overall, the analysis seems reasonable. However, some important claims have been misrepresented or are at the very least questionable, which belies the contributions of the paper. I believe that important parts of the analysis of CAESAR have been ignored or abstracted away, that may significantly increase the sample and computational complexity. In particular:

- the sample complexity of coarse visitation estimation for all policies.

- the cost of solving the optimization problem for the sampling distribution in Eq. (5)

- analysis of the weight estimation algorithm (IDES)


In addition, I feel that comparisons to the most relevant works are inaccurate, in particular:

- claimed differences with Amortila et al. (2024), e.g., that CAESAR's objective is easier to solve (L144-154, left)

- the claimed novelty of the coarse visitation estimation results (Section 4.1)

**Sample complexity of estimating visitation for all policies: dependencies on state space $S$ and policies $K$**

The abstract states that $\tilde O(\varepsilon^{-1})$ samples are required to estimate the visitation distributions of all policies, which is required in step 1 of CAESAR (Algorithm 2). Correct me if I'm wrong, but I believe that the sample complexity of step 1 is actually something like
$$O\left(\frac{K S A}{\varepsilon^{-1}}\right).$$
As we often treat $S$ (and seemingly $K$ here) as being very large (potentially exponentially), this is not a lower-order term to be absorbed into the total sample complexity of CAESAR (Theorem 4.9) without further assumptions. None of this has been clearly stated in the main body nor discussed, and I believe that it warrants extended consideration as it does form a trade-off.

I am also concerned that a large number of additional samples will be required to compute (5); this computed distribution is the foundation of the paper but has been completely abstracted away, see below.

**Optimization problem seems expensive and difficult to solve**

Since the sampling distribution computed from the optimization problem in (5) is central to the paper, I think it's necessary to provide evidence that it can be solved, and under what conditions it can be solved. I am concerned about this because
- The inner maximization is non-concave as it is over a discrete set, and relaxations to convex hulls might incur larger sample complexities
- The entire paper of Amortila et al. (2024) is about methods to solve (5), and I am not sure that any of them apply without relaxations of the objective and model access / online rollouts. Therefore the expense to solve (5) may dominate the total sample complexity of CAESAR, or fall outside the claimed problem setup (only one data collection phase). In addition, the the authors should probably discuss the robustness of the downstream methods (IDES etc) in the case that $\hat \mu^*$ is inaccurate.


**Analysis of the weight estimation method (IDES)**

The gradient is estimated using samples from $\mu$, but the sample complexity of estimating this gradient is not considered and is assumed away (as far as I can tell in the main body, since Lemma 4.6 simply assumes that we have an accurate $\hat w$). It seems probable that this will incur worse dependencies in the total sample complexity of CAESAR.

**Comparisons to Amortila et al. (2024)**

In L144-154 (left), the authors claim that the optimization problem in Eq. (5) is easier to solve than the one in Amortila et al. (2024); I think this simply may not be true given that the inner maximization in Eq. (5) is over a non-concave / discrete set, nor do the authors actually discuss how to solve Eq. (5). Also in general, stating that Amortila et al. (2024) is a "concurrent work" (L144) is simply dishonest, as it precedes this submission by 1 year, and deserves much greater credit in Section 4.2.

**Novelty of coarse visitation estimation**

Zhang and Zanette (2023), which is not cited in the submission, has already demonstrated how to estimate visitations up to constant accuracy, and their Lemma A.22 is essentially the same result as Eq. (1). Although their setting is different, it's related in the sense that they use coarse estimation to get a fast rate and use coarsely estimated occupancies to explore. I think it is not fair to claim coarse estimation as a contribution in this submission, without at least heavily citing Zhang an Zanette (2023).

**References**

Zhang, Ruiqi, and Andrea Zanette. "Policy finetuning in reinforcement learning via design of experiments using offline data." Advances in Neural Information Processing Systems 36 (2023): 59953-59995.

**Essential References Not Discussed:**

Important references are missing and/or inadequately credited
- As mentioned previously, Zhang and Zanette (2023) show how to coarsely estimate distributions, and show how to use them for exploration. Another highly relevant work along this vein is Li et al. (2023), who also estimate distributions up to constant accuracy, then use them to compute a covering distribution (for tabular MDPs, but they describe how to optimize the objective efficiently).
- The optimization problem in Eq. (5) is identical to the one in Amortila et al. (2024)
- Techniques behind gradient-based methods to estimate weights, and handling error propagation of estimated weights and visitations, have been utilized in in Huang and Jiang 2024; Kallus and Uehara 2020; Amortila et al. 2024; Huang et al. 2024

**References**

Li, G., Zhan, W., Lee, J. D., Chi, Y., & Chen, Y. (2023). Reward-agnostic fine-tuning: Provable statistical benefits of hybrid reinforcement learning. Advances in Neural Information Processing Systems, 36, 55582-55615.

Kallus, N., & Uehara, M. (2020, November). Statistically efficient off-policy policy gradients. In International Conference on Machine Learning (pp. 5089-5100). PMLR.

Huang, A., & Jiang, N. (2024). Occupancy-based Policy Gradient: Estimation, Convergence, and Optimality. Advances in Neural Information Processing Systems, 37, 416-468.

Amortila, P., Foster, D. J., Jiang, N., Sekhari, A., & Xie, T. (2024). Harnessing density ratios for online reinforcement learning. arXiv preprint arXiv:2401.09681.

Huang, A., Chen, J., & Jiang, N. (2023, July). Reinforcement learning in low-rank mdps with density features. In International Conference on Machine Learning (pp. 13710-13752). PMLR.

**Experimental Designs Or Analyses:**

There are no experiments.

**Methods And Evaluation Criteria:**

As this work is theoretical in nature, I would say that the guarantees the authors have desired to prove make sense for the stated problem.

**Other Comments Or Suggestions:**

n/a

**Other Strengths And Weaknesses:**

I think that the technical contributions and conceptual insights of this paper are limited in novelty, in the sense that they are largely derived from existing papers and combined for the task of multiple policy off-policy evaluation.

However, the observations that (a) one only needs to estimate the visitation distributions up to constant accuracy and (b) one can extract useful importance weights from this seem interesting and useful. At the same time, all of this relies on being able to solve Eq. (5), which to me, is the hardest part of the problem.

I am recommending rejection because of the obfuscated presentation of results (sample complexity), the lack of consideration towards how to solve Eq. (5), and the improper acknowledgement of existing work from which many technical tools were almost surely derived.

**Questions For Authors:**

1. What is the sample complexity of step 1 of CAESAR, i.e., estimating the visitations of all policies?
2. How can Eq. (5) be solved, and can you comment on the expected sample complexity?
3. Since the gradient in IDES is estimated, can you comment on the expected complexity of learning $\hat w$ via SGD?

**Relation To Broader Scientific Literature:**

The paper combines methods from previous literature on weight and policy visitation estimation, and reward-free exploration with visitations and weight estimation, to the setting of multiple policy off-policy evaluation. The goal is to reduce linear dependence in $K$ to a better coverage coefficient.

**Theoretical Claims:**

I did not check the proofs, but the high-level dependencies in the theoretical claims seem consistent with what one would expect and with those in existing work.

---

> ### Author Rebuttal · Authors · 2025-03-30
>
> Thanks the reviewer for the detailed comments on our work. We are confident that we can solve your concerns well. And we kindly request you to consider increasing your score based on our following elaboration.
>
> One of your main concerns is about our optimization objective (5):
> > The problem (5) is hard to solve since the inner maximization is non-concave. The sample complexity of solving (5) is unclear.
>
> We want to correct this comment since it is not true. Our opt objective is a well-formulated convex problem. Let's denote $\sum_{s,a} \frac{(\hat d_h^{\pi^k}(s,a))^2}{\mu_h(s,a)}$ by $f_k(\mu)$. It is trivial that $f_k(\mu)$ is convex w.r.t $\mu$, and we know the maximum of a bunch of convex functions is still convex, hence, our objective $\max_{k\in[K]} f_k(\mu)$ is convex. It is ready to be solved by any convex optimization solver which means no sample is needed in solving (5).
>
> > Our opt problem (5) is identical to the one in Amortila et al. (2024).
>
> The formulation same to the one in Amortila et al. (2024) is (4) instead of (5). (4) is common in many works since it is a standard formulation to describe the coverage property of a dataset. The emphasis of two works is different. We show that an up to constant optimal $\mu$ is enough in our scenario, hence we use coarse estimators to approximate (4), leading us to (5) which is easy to solve. They want an exact optimizer to (4) which is not straightforward to solve since the true visitation distribution is unknown, and they have to reformulate the problem as a RL problem.
>
> > The sample complexity of coarse estimation for all policies depends on poly(K).
>
> It is not true. We want to emphasize that our low order term for coarse estimation is always bounded by $\tilde{O}(poly(H,S,A)/\epsilon)$ without $poly(K)$ which is a non-trivial contribution. We achieve this by proposing MARCH algorithm along with novel theoretical analysis (utilizing our proposed $\beta$-distance which possesses lots of nice properties). We briefly discussed it in Line 240-249 (left) in the main page. The detailed content is in Appendix B due to page limits. We will consider updating the paper to emphasize it more in the main page.
>
> > The complexity of estimating $\hat w$ in IDES is unclear.
>
> The sample complexity of IDES is exactly the number of optimization iterations since we are using stochastic gradient. Specifically, in each iteration of optimization, one sample is used to estimate the stochastic gradient. The sample complexity of IDES is same as the one presented in Theorem 4.9, which is also presented in Algorithm 1 where the iteration number $n_h$ is specified.
>
> > Missing related works.
>
> We appreciate it that you listed these related works which we didn't mention in the current version carelessly due to the massive amount of literature. We will definitely update our paper to include these works as well as the discussions on relations.
>
> We also want to clarify here that the statement of "Amortila et al. (2024) is a "concurrent work" (L144)" is not dishonest. Our first draft version was made public in the same period as Amortila et al. (2024). But we understand reviewers have no information on our submission. We will update the expression to make it more appropriate for this submission.
>
> > The novelty of coarse estimation and other derivation techniques is limited.
>
> We agree that low order sub-procedure is not new in the literature. And thanks for providing these related works. We will definitely include them in the updated version. However, we disagree that the existing results can imply our results on coarse estimation. The Lemma A.22 in Zhang and Zanette (2023) is based on estimation of transition functions, while we directly estimate visitations. And in our result, the constant (i.e. $c$ in Definition 4.1) can be any value (see Appendix A). Eq (45) in Li et al. (2023) is similar to our results,  however, our formulation is much cleaner and more elegant based on simple concentration inequalities while they have more additional complex terms. We formally formulate the technique as coarse estimation in this work and present results that are ready to use in other different scenarios.
>
> We disagree with the comment that the contribution of our work is just simply combining existing techniques. We presented our contributions clearly in the end of introduction section. Multiple policy evaluation is a very challenging and under-studied problem, and we provide very nice theoretical results for this problem which is a significant step along with many useful side products which have many potential usages.
>
> Again, we appreciate your efforts on reviewing our paper. Please let us know if our above rebuttal solves your concerns and we are happy to keep discussing if you have more questions.

---

### Decision · Program_Chairs · 2025-05-01

**Decision:**

Accept (poster)

**Comment:**

The reviewers have raised several concerns, and the authors have addressed most of them. The paper has strengths and weaknesses:

**Strengths**

- Clarity: Well-organized, easy-to-follow proofs and a clear explanation of CAESAR.

- Novelty: Introduces IDES and MARCH, advancing offline policy evaluation, with results that do not depend on the number of policies.

**Weaknesses**

- No Empirical Validation: Lacks experiments to support the practical effectiveness.

- Coarse Estimation: The coarse estimation concept requires further justification and comparison with existing works.

Overall, although the paper has received mixed scores, I am more inclined towards acceptance.